# The role of subtemperate slip in thermally-driven ice stream margin migration

Marianne Haseloff[1,2], Christian Schoof[2], and Olivier Gagliardini[3]

[1]Atmospheric and Oceanic Sciences Program, Princeton University, Princeton, NJ 08540, USA
[2]Department of Earth, Ocean and Atmospheric Sciences, The University of British Columbia, Vancouver, BC V6T 1Z4, Canada
[3]Univ. Grenoble Alpes, CNRS, IRD, Grenoble INP, IGE, 38000 Grenoble, France

*Correspondence to:* Marianne Haseloff (marianne.haseloff@earth.ox.ac.uk)

**Abstract.** The amount of ice discharged by an ice stream depends on its width, and the widths of unconfined ice streams such as the Siple Coast ice streams in West Antarctica have been observed to evolve on decadal to centennial timescales. Thermally-driven widening of ice streams provides a mechanism for this observed variability through melting of the frozen beds of adjacent ice ridges. This widening is driven by the heat dissipation in the ice stream margin, where strain rates are high, and at the bed of the ice ridge, where subtemperate sliding is possible. The inflow of cold ice from the neighboring ice ridges impedes ice stream widening. Determining the migration rate of the margin requires resolving conductive and advective heat transfer processes on very small scales in the ice stream margin, and these processes cannot be resolved by large scale ice sheet models. Here, we exploit the thermal boundary layer structure in the ice stream margin to investigate how the migration rate depends on these different processes. We derive a parameterization of the migration rate in terms of parameters that can be estimated from observations or large scale model outputs, including the lateral shear stress in the ice stream margin, the ice thickness of the stream, the influx of ice from the ridge, and the bed temperature of the ice ridge. This parameterization will allow the incorporation of ice stream margin migration into large-scale ice sheet models.

## 1 Introduction

The Siple Coast ice streams are fast moving regions within the West Antarctic ice sheet. They exhibit temporal changes on decadal to centennial time scales in their spatial configuration, for example slow down and reactivation cycles and changes in ice stream width (Stephenson and Bindschadler, 1988; Retzlaff and Bentley, 1993; Harrison et al., 1998; Hamilton et al., 1998; Echelmeyer and Harrison, 1999; Fahnestock et al., 2000; Conway et al., 2002; Catania et al., 2006; Catania et al., 2012; Bindschadler et al., 2000; Stearns et al., 2005; Hulbe and Fahnestock, 2007). The widening and narrowing of ice streams can strongly affect mass discharge from an ice stream: simplified ice stream models show that the ice stream velocity and discharge strongly increase with stream width (Raymond, 2000). Correctly modeling the evolution of ice streams, including the migration of their margins, is therefore essential for reliable predictions of the evolution of the Antarctic ice sheet (Bamber et al., 2000)

Ice streams are bordered by slowly moving ice, called ice ridges, and the close proximity of fast to slowly moving ice is reflected in a sharp gradient in basal resistance between ridge and stream (Bentley et al., 1998). The question of margin

migration is tightly linked to the question of how this gradient is sustained. In the absence of freezing in the bed, subglacial drainage can in principle widen ice streams, if water transport is in the direction of effective pressure gradients (Haseloff, 2015). In this scenario, infinite widening can be suppressed by the formation of ice ridges (e.g. in Kyrke-Smith et al., 2014). This leads to a gradient in ice overburden pressure that counteracts gradients in effective pressure, so that there is no net water

pressure gradient driving flow of water towards the ridge. Alternatively, Perol et al. (2015) propose the existence of a channel co-located with the ice stream margin, which theoretically locks the position of the margin into place: margin migration now requires a reorganization of the subglacial drainage system.

However, if freezing in the bed is possible, a thermal barrier can form in the bed which suppresses widening through subglacial drainage (Haseloff, 2015). The existence of such a thermal barrier is supported by radar observations under some

ice streams and ridges, where strong contrasts in basal reflectivity from stream to ridge have been interpreted as transitions from a temperate to a frozen bed (Bentley et al., 1998; Catania et al., 2003). Under these conditions, the inwards migration (or narrowing) of an ice stream requires freezing of the entire sediment column (Schoof, 2012, appendix B). As melt water can be supplied to sections of the bed with active freezing from other regions of the ice stream via subglacial drainage, this necessitates taking into account the ice-stream-wide energy balance. Consequently, the inwards migration of ice streams is the

result of insufficient heat dissipation over the width of the entire ice stream (Haseloff, 2015). However, as shown in Haseloff (2015) this process can at least in principle be modelled with large-scale ice sheet models without recourse to a boundary layer.

In this scenario the outwards migration of ice stream margins requires melting of the frozen sediment under the ice ridge. By contrast with a narrowing ice stream, it is however not necessary for the entire thickness of the sediment column to melt out: only part of it needs to be unfrozen to permit sliding, and we will later idealize this by assuming that sliding is possible as

soon as the melting point is reached at the bed. This however also underlines the asymmetry between widening and narrowing of an ice stream, which motivates us to focus on the harder problem of widening, which requires heat to be transferred into the bed. Several studies show that a strong gradient in basal resistance created by a thermal transition leads to significant englacial heat production in the ice stream margins (Raymond, 1996; Jacobson and Raymond, 1998; Schoof, 2006; Suckale et al., 2014; Perol and Rice, 2015). Combined with conductive heat transfer, this heat dissipation can lead to the outwards migration of the

margins by warming the bed outside the active ice stream (Schoof, 2012; Haseloff et al., 2015). This migration is counteracted by advective cooling through the inflow of ice from the ice ridge, driven by an elevation difference between ice ridge and ice stream. The rate of migration is highly sensitive to the relative strength of these two processes (Jacobson and Raymond, 1998; Haseloff et al., 2015).

Existing studies that derive a migration rate from this competition between dissipation, conduction, and advection (Schoof,

2012; Haseloff et al., 2015) assume that the transition from a temperate to a frozen bed is co-located with an abrupt transition from free slip to no slip. However, it is unlikely that such a transition occurs in reality: the basal shear stress goes to infinity at a no-slip to free-slip transition. Therefore a no slip boundary condition on the cold side requires slip to be suppressed there for any amount of basal shear stress (Fowler, 2013). Instead, we expect sliding to occur, either due to mechanical failure or due to a residual premelted water film at the ice–bed contact (Fowler, 1986; Echelmeyer and Zhongxiang, 1987; Cuffey et al., 1999;

Schoof, 2004; Platt et al., 2016; Elsworth and Suckale, 2016). Both of these processes would lead to subtemperate sliding, that

is, sliding at temperatures below the melting point. Additionally, the high stress concentrations may be alleviated by mechanical failure or damage production in the ice itself (Pralong and Funk, 2005).

In the presence of subtemperate slip, we expect significant changes to the velocity field, which is responsible for advection of heat, and to the spatial distribution of heat dissipation. In particular, heat is then dissipated at the frozen ice–bed interface. This is the very location where warming has to occur in order for the ice stream margin to migrate outwards. We therefore expect subtemperate slip to have a significant influence on the rate at which ice stream margins can migrate.

To determine the rate of margin migration, we have to consider the thermal and mechanical transitions from ice ridge to ice stream flow, which take place over a distance of just a few ice thicknesses. This is narrow in comparison to the width of the ice ridge and the ice stream, and can be captured by a boundary layer model (Haseloff et al., 2015). The physics captured by the boundary layer model is not necessarily included in large scale ice sheet models, and requires very high resolution of the computational grid (Haseloff et al., 2015). The purpose of this paper is therefore twofold: $(i)$ to use the margin boundary layer model of Haseloff et al. (2015) to investigate how subtemperate slip changes the heat production and temperature field in the ice stream margin, and thereby the rate at which ice streams can migrate outwards and $(ii)$ to derive parameterizations of the margin migration rate which can be used in large scale ice sheet models. Both of these points go beyond the work in Haseloff et al. (2015): the parameterizations we derive in particular show how the limit of rapid advection of heat across the shear margin can be used to simplify the boundary layer model and arrive at tractable forms of the migration rate that could be implemented in computational models either in the form of semi-analytical formulae or lookup tables.

This paper is laid out as follows: we state the model in section 2. Typical solutions of the model are presented in section 3, where we also explain how we determine the migration velocity. The dependence of the migration velocity on forcing parameters is determined in section 4, and in sections 4.3–4.4 we derive a parameterization of the migration velocity as function of these forcing parameters. We discuss our results in section 5.

## 2  The model

The model for ice stream margins we use here is derived in Haseloff et al. (2015). Let $(x, y', z)$ be a fixed coordinate system. The model assumes a well-developed ice stream, whose principal flow direction is aligned with the positive $x$–direction, as shown in figure 1a. The $y'$-axis is transverse to the ice stream, and the $z$-axis is vertical. The ice stream is bordered by slowly moving ice ridges. The model for the ice stream margin is located at the transition from ice stream flow regime to ice ridge flow regime, providing the coupling between the two.

In contrast to typical 'shallow' ice stream and ice sheet models (Fowler and Larson, 1978; Morland and Johnson, 1980; Hutter, 1983; Muszynski and Birchfield, 1987; MacAyeal, 1989; Blatter, 1995; Pattyn, 2003), which assume a small aspect ratio between vertical and lateral extent, the width of the margin region captured by the boundary layer model is of the order of the ice stream thickness. Consequently, the far fields of the ice ridge and ice stream are attained at $y \pm \infty$ (see figure 1b).

The asymptotic analysis in Haseloff et al. (2015) shows that the boundary layer evolves rapidly in comparison to the ice stream and ice ridge, and is consequently quasi-static with the only time-dependence arising from the moving transition be-

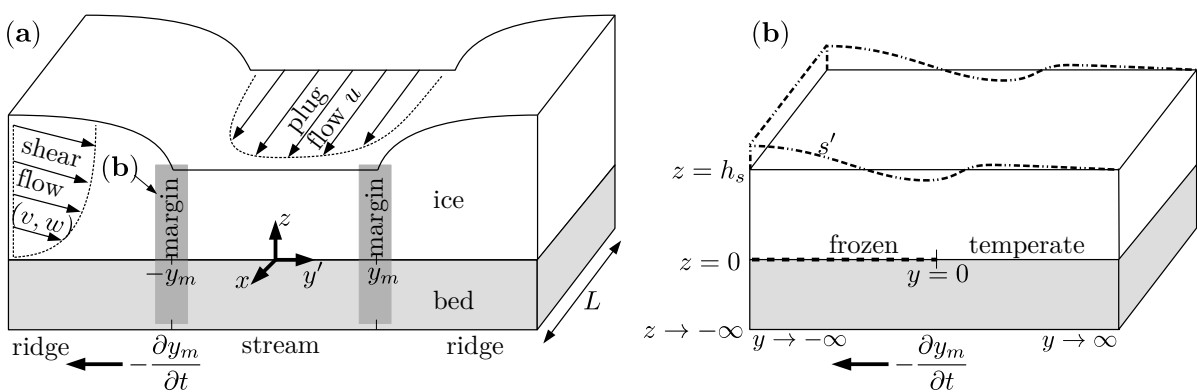

**Figure 1.** Panel a: Large scale model geometry: we assume an ice stream flowing in positive $x$–direction. At its sides, the ice stream is bordered by slowly moving ice ridges. Panel b: boundary layer geometry, which moves at the rate $v_m = \partial y_m / \partial t$ through the ice stream/ridge geometry shown in panel a.

tween a frozen and a temperate bed at $\pm y_m(x,t)$. Morever, Haseloff et al. (2015) show that the surface of the ice stream margin is flat at leading order and located at $z = h_s$, where $h_s$ is the ice thickness of the ice stream.

The ice-bed interface is assumed to be flat and located at $z = 0$. Note however that this assumption does not preclude the application of our results to ice streams with a weak topographic control, as found in many regions of the Siple Coast: this assumption merely requires the elevation gradient of the bed to be sufficiently small that the bed elevation does not vary significantly over lateral distance of a few ice thicknesses.

We define the margin location $y' = -y_m(x,t)$ as the point where the bed goes from being at the melting point to below the melting point. To facilitate our analysis, we immediately change to a moving coordinate frame in which the transition from a temperate to a frozen bed is stationary at $y = 0$, see figure 1b. In other words, if $y'$ is the stationary coordinate with $y' = 0$ in the ice stream center and the margin at $y' = -y_m(x,t)$, then the lateral coordinate $y$ of the margin model is linked to $y'$ through

$$y = y' + y_m(x,t), \tag{1}$$

with $y = 0$ at the transition from frozen to temperate bed (see figure 1b). $y$ increases towards the ice stream, with negative $y$ corresponding to the ice ridge side of the domain. The boundary layer is moving at velocity $-\partial y_m / \partial t$ with respect to the fixed $y'$-axis.

The boundary layer is effectively two-dimensional: the ice stream is much longer than a single ice thickness, and therefore along-flow variations in mechanical and thermal conditions in the $x$-direction are much smaller than corresponding variations in the transverse and vertical directions. In other words, the $x$-coordinate is passive and $(y, z)$ are the independent variables in the model.

We assume that the thermal state of the ice-bed interface controls the basal boundary conditions for the ice. This requires us to model the thermal response not only of the ice but also of the bed. We therefore specifically include the bed in the domain, and apply a geothermal heat flux at $z \to -\infty$. At the lateral domain boundaries at $y \pm \infty$, far field boundary conditions are determined by coupling with stream and ridge.

Force balance can be separated into a downstream component, with $u$ denoting the velocity component in the $x$–direction, and a transverse component in the $(y, z)$-plane, with $(v, w)$ denoting the corresponding transverse velocity plane. In the downstream direction, the velocity $u$ is determined by

$$\frac{\partial}{\partial y}\left(\eta\frac{\partial u}{\partial y}\right) + \frac{\partial}{\partial z}\left(\eta\frac{\partial u}{\partial z}\right) = 0 \tag{2}$$

with $\eta$ the viscosity. The transverse velocity field is determined by the two-dimensional Stokes and mass balance equations

$$\frac{\partial}{\partial y}\left(2\eta\frac{\partial v}{\partial y}\right) + \frac{\partial}{\partial z}\left[\eta\left(\frac{\partial v}{\partial z} + \frac{\partial w}{\partial y}\right)\right] - \frac{\partial p}{\partial y} = 0, \tag{3a}$$

$$\frac{\partial}{\partial y}\left[\eta\left(\frac{\partial v}{\partial z} + \frac{\partial w}{\partial y}\right)\right] + \frac{\partial}{\partial z}\left(2\eta\frac{\partial w}{\partial z}\right) - \frac{\partial p}{\partial z} = 0, \tag{3b}$$

$$\frac{\partial v}{\partial y} + \frac{\partial w}{\partial z} = 0. \tag{3c}$$

The viscosity $\eta$ depends on all three velocity components through Glen's law (Paterson, 1994)

$$\eta = \frac{A^{-1/n}}{2^{1/n}}\left[\left|\frac{\partial u}{\partial y}\right|^2 + \left|\frac{\partial u}{\partial z}\right|^2 + \left|\frac{\partial v}{\partial z} + \frac{\partial w}{\partial y}\right|^2 + 2\left|\frac{\partial v}{\partial y}\right|^2 + 2\left|\frac{\partial w}{\partial z}\right|^2\right]^{\frac{1-n}{2n}} \tag{4}$$

with $A$ the usual viscosity parameter and $n$ the rheology exponent. For simplicity, we neglect the effect of temperature on viscosity here.

     The ice stream imposes a lateral shear stress $\tau_s$ as a far-field boundary condition. Additionally, the plug flow in the ice stream requires a vertically-uniform across-stream velocity in this far-field, so

$$\eta\frac{\partial u}{\partial y} \to \tau_s, \qquad \frac{\partial v}{\partial z} \to 0, \qquad w \to 0 \qquad \text{for } y \to \infty. \tag{5}$$

Towards the ice ridge, we expect a shearing flow in the transverse direction and negligible flow in the downstream direction

$$\left.\begin{aligned} u &\to 0 \\ v &\to \frac{(n+2)}{(n+1)}\frac{q_r}{h_s}\left[1 - \left(1 - \frac{z}{h_s}\right)^{n+1}\right] \\ w &\to 0 \end{aligned}\right\} \qquad \text{for } y \to -\infty \tag{6}$$

with $q_r$ the ice flux from the ice ridge towards the ice stream and $h_s$ the ice thickness of the ice stream; the form of $v$ corresponds to a 'shallow ice'-type shearing flow with a temperature-independent rate factor $A$.

     We assume that basal melting has a negligible effect on ice velocities, so

$w = 0$   at $z = 0$.      (7)

On the temperate (stream-ward) side of the ice–bed interface, we assume that the basal shear stress is negligible compared with the shear stresses in the ice, leading to a free slip boundary condition:

$$\eta \frac{\partial u}{\partial z} = \eta \frac{\partial v}{\partial z} = 0 \quad \text{for } y > 0,\, z = 0. \tag{8}$$

In posing this boundary condition for an ice stream that is actively widening, we are assuming that an infinitesimal amount of melting of the bed suffices to allow for slip: once the thermal barrier at the bed is breached, we only need a very thin ice-free layer in order for slip to occur. This is consistent at least with the idea of a plastic bed, where slip can happen on a plane, or with a hard bed.

To the extent that additional degrees of freedom (other than temperature) are involved in sliding, the main concern would presumably be water pressure at the bed or within the till, rather than the thickness of the unfrozen till layer. Our assumption of a free slip once the melting point is reached is best justified (see Haseloff et al, 2015) if we suppose that the unfrozen bed is hydraulically well-connected, so that the water pressure in the parts of the bed that have just become unfrozen quickly equilibrates with water pressure elsewhere under the ice stream (and hence basal friction is comparable to the rest of the active ice stream). Shear stresses experienced by the margins of the ice stream are large compared with basal drag throughout the ice stream (Haseloff et al, 2015), and this implies that basal friction is small at leading order everywhere where the melting point is reached. There are undoubtedly other, more elaborate models for basal shear stress of the unfrozen bed; ours is the simplest possible case to analyse.

Where the bed is frozen, we consider two different possibilities. The first assumes that no slip is possible, so that the basal boundary condition for $y < 0$ is

$$u = v = 0 \quad \text{for } y < 0,\, z = 0. \tag{9a}$$

We also investigate the possibility of slip at significant basal friction on the frozen bed. For simplicity we use the simplest possible version of this problem and assume that the frozen ice-bed contact fails at a fixed yield stress $\tau_c$ (Schoof, 2004, 2010)

$$\left. \begin{array}{ll} \text{either} & \eta \frac{\partial u}{\partial z} = \tau_c \frac{u}{\sqrt{u^2+v^2}}, \quad \eta \frac{\partial v}{\partial z} = \tau_c \frac{v}{\sqrt{u^2+v^2}}, \quad \sqrt{u^2+v^2} > 0 \\ \text{or} & \sqrt{\left(\eta \frac{\partial u}{\partial z}\right)^2 + \left(\eta \frac{\partial v}{\partial z}\right)^2} \leq \tau_c, \quad \sqrt{u^2+v^2} = 0 \end{array} \right\} \text{for } y < 0,\, z = 0. \tag{9b}$$

The no-slip case (9a) can be obtained formally by putting $\tau_c = \infty$ in (9b).

The upper surface is traction-free, and flat at leading order. In practice, this implies vanishing shear stress and normal velocity, with vanishing normal stress accounted for by a first order correction to the constant leading order surface elevation. If the actual upper surface is located at $h_s + s'$ with $s' \ll h_s$, then

$$\eta \frac{\partial u}{\partial z} = \eta \frac{\partial v}{\partial z} = w = 0, \quad 2\eta \frac{\partial w}{\partial z} - p + \rho g s' = 0 \quad \text{at } z = h_s. \tag{10}$$

Thus, even though our model geometry is a parallel-sided strip, it takes into account the first order surface slope towards the ice ridge.

Note that we have formulated the flow problem in such a way that it can be solved without reference to the temperature field. Physically, we however require that the temperature $T$ is below the melting point $T_m$ for $y < 0$, $z = 0$ and that the temperature

is at the melting point for $y \geq 0$, $z = 0$. To ensure that these conditions are met, we have to solve the heat equation in the ice $(0 < z < h_s)$ and in the bed $(z < 0)$. The englacial heat production in the ice is balanced by conductive and advective heat transport, as well as a pseudo-advective term which is the result of the ice stream margin migrating at the rate

$$v_m := \frac{\partial y_m}{\partial t} \tag{11}$$

into the ice ridge. (Physically, this term represents the effect of having to warm the initially cold ice in the ice ridge as the margin migrates into the ridge.) In the bed, no heat is dissipated and the bed is assumed to be static, so that we have a balance between the same pseudo-advective term and diffusion of heat:

$$\rho c_p \left( v_m \frac{\partial T}{\partial y} + v \frac{\partial T}{\partial y} + w \frac{\partial T}{\partial z} \right) - k \left( \frac{\partial^2 T}{\partial y^2} + \frac{\partial^2 T}{\partial z^2} \right) = a \quad \text{for } 0 < z < h_s, \tag{12a}$$

$$\rho_{\text{bed}} c_{p,\text{bed}} v_m \frac{\partial T}{\partial y} - k_{\text{bed}} \left( \frac{\partial^2 T}{\partial y^2} + \frac{\partial^2 T}{\partial z^2} \right) = 0 \quad \text{for } z < 0 \tag{12b}$$

where we used the margin migration velocity $v_m$ as defined by (11). $\rho$ and $\rho_{\text{bed}}$ are the densities of ice and bed, respectively, $c_p$ and $c_{p,\text{bed}}$ are specific heat capacities, and $k$ and $k_{\text{bed}}$ are thermal conductivities (see table 1). The heat production term $a$ provides the thermo-mechanical coupling:

$$a = \frac{A^{-1/n}}{2^{1/n}} \left( \left| \frac{\partial u}{\partial y} \right|^2 + \left| \frac{\partial u}{\partial z} \right|^2 + \left| \frac{\partial v}{\partial z} + \frac{\partial w}{\partial y} \right|^2 + 2 \left| \frac{\partial v}{\partial y} \right|^2 + 2 \left| \frac{\partial w}{\partial z} \right|^2 \right)^{\frac{1+n}{2n}}. \tag{13}$$

Advection from the ice ridge prescribes a far-field temperature profile $T_r(z)$, while there are no significant lateral temperature gradients towards the ice stream far field

$$T = T_r(z) \qquad \text{for } y \to -\infty, \qquad\qquad \frac{\partial T}{\partial y} \to 0 \qquad \text{for } y \to \infty. \tag{14}$$

To be consistent with a conduction-dominated temperature field, we assume for the far field ridge temperature

$$T_r = \begin{cases} T_s + \dfrac{q_{\text{geo}}}{k} (h_s - z) & \text{if } z \geq 0 \\[2mm] T_s + \dfrac{q_{\text{geo}}}{k} \left( h_s - \dfrac{k}{k_{\text{bed}}} z \right) & \text{if } z < 0. \end{cases} \tag{15}$$

We will show in section 4 that the migration velocity is sensitive only to the far field temperature at the bed, so the specific form of $T_r$ is immaterial. Here we have assumed a linear profile for simplicity.

At the surface at $z = h_s$, we assume a constant surface temperature $T_s$, and towards $z \to -\infty$ we assume a constant geothermal heat flux $q_{\text{geo}}$

$$T = T_s \qquad \text{at } z = h_s, \qquad\qquad -k_{\text{bed}} \frac{\partial T}{\partial z} \to q_{\text{geo}} \qquad \text{as } z \to -\infty. \tag{16}$$

Finally, at the bed, we impose the following boundary conditions and inequality constraints:

$$T < T_m \quad \text{and} \quad -k \left.\frac{\partial T}{\partial z}\right|^+ + k_{\text{bed}} \left.\frac{\partial T}{\partial z}\right|^- = \begin{cases} 0 & \text{if } \tau_c = \infty \\ \tau_c \sqrt{u^2 + v^2} & \text{if } \tau_c < \infty \end{cases} \quad \text{for } y < 0,\ z = 0, \tag{17a}$$

$$T = T_m \quad \text{and} \quad -k \left.\frac{\partial T}{\partial z}\right|^+ + k_{\text{bed}} \left.\frac{\partial T}{\partial z}\right|^- < \begin{cases} 0 & \text{if } \tau_c = \infty \\ \infty & \text{if } \tau_c < \infty \end{cases} \quad \text{for } y > 0,\ z = 0. \tag{17b}$$

The two cases in $\tau_c$ correspond to whether subtemperate slip is or is not possible ($\tau_c$ finite or infinite, respectively).

The equalities in (17) arise from the construction of our model: we have chosen the location $y = 0$ to separate a region with a temperate bed ($y > 0$) from one where the bed temperature must be below the melting point, but where it is not otherwise prescribed. In the latter case, a flux condition is necessary to ensure conservation of energy at the bed. Consequently, the temperature inequality in (17a) is an intrinsic part of how we have defined our domain, with the transition from a subtemperate to temperate bed occurring at $y = 0$ in our traveling coordinate system.

The flux constraint in (17b) by contrast is really a constraint on freezing rates on the temperate side of the thermal transition at the bed, and in imposing it we are assuming that the margin is migrating into the ice ridge at a rate $v_m > 0$. A local analysis of the temperature field near $y = 0$ (see appendix A for a summary and the supplementary material, section S2 and Schoof (2012) for details) demonstrates that, if the temperature constraint in $(17a)_1$ is satisfied, then the net heat flux out of the bed for small $y > 0$ either approaches $+\infty$ or equals the basal dissipation rate attained at small negative $y < 0$. When the margin

migrates towards the ice ridge, we assume an infinite basal freezing rate cannot occur on the temperate side of the thermal transition at $y = 0$, $z = 0$, leaving only the possibility of a finite heat flux (the second version of the inequality in $(17b)_2$). For the case of no subtemperate slip, this corresponds neatly to having no freezing at the bed at all on the temperate side of the transition (see also Haseloff et al., 2015; Schoof, 2012). For the case of subtemperate slip, our formulation of having an abrupt transition from a finite value of $\tau_c$ to free slip implies a discontinuity in basal heat production, and consequently

requires a finite but non-zero basal freezing rate near the origin. In order to maintain the bed at the melting point, that finite freezing rate has to be compensated for by subglacial drainage (that is, a finite supply of latent heat into the very tip of the temperate bed region). We anticipate that future versions of this model will consider smooth transitions from subtemperate to temperate sliding, for instance by allowing the yield stress to approach zero continuously as function of $T$. This however is computationally extremely onerous, and we persist with our simpler version of the basal physics here.

The inequality constraints serve the role of determining a unique migration rate $v_m$. If we were to dispense with them, we could solve (12a)–(16) with an arbitrary choice of $v_m$. However, for fixed model parameters, an arbitrary choice of $v_m$ will see one of the inequality constraints in (17) violated, and their role is therefore to specify the migration rate (see supplementary material, section S1, and Schoof (2012)). That migration rate is then a function of geometrical and forcing parameters such as $h_s$, $\tau_s$, $q_r$, and $q_{\text{geo}}$. $\tau_s$ and $q_r$ in particular represent the far field forcing due to coupling with ice stream and ice ridge.

For instance, from the perspective of the ice stream, $\tau_s$ is the lateral shear stress it imposes on the boundary, while from the perspective of the ice ridge, $q_r$ is the rate at which it supplies mass to the ice stream through the margin.

**Table 1.** Parameter values used in the sample calculations presented here. Ice stream thickness $h_s$, lateral inflow of ice from the ridge $q_r$ and marginal lateral shear stress $\tau_s$ are highlighted as they represent coupling with ice ridge and ice stream dynamics. $h_s$ and $\tau_s$ correspond to the values observed at the upper margin of Whillans ice stream (Harrison et al., 1998), which migrates at a rate of 7 to 30 m year$^{-1}$ (Hamilton et al., 1998; Harrison et al., 1998; Echelmeyer and Harrison, 1999). The $q_r$ estimate is based on an inflow velocity of 10 m year$^{-1}$.

| Description | Symbol | Value | Units |
|---|---|---|---|
| viscosity parameter | $A$ | $1.6 \times 10^{-15}$ | kPa$^{-3}$ s$^{-1}$ |
| rheological exponent | $n$ | 3 | |
| specific heat capacity | $c_p$, $c_{p,\text{bed}}$ | 2 | kJ kg$^{-1}$ K$^{-1}$ |
| acceleration due to gravity | $g$ | 9.81 | m s$^{-2}$ |
| thermal conductivity | $k$, $k_{\text{bed}}$ | 2.3 | W m$^{-1}$ K$^{-1}$ |
| geothermal heat flux | $q_{\text{geo}}$ | $6 \times 10^{-2}$ | W m$^{-2}$ |
| density of ice | $\rho$ | 920 | kg m$^{-3}$ |
| density of bed | $\rho_{\text{bed}}$ | 920 | kg m$^{-3}$ |
| surface temperature | $T_s$ | $-25$ | $^o$C |
| melting point | $T_m$ | 0 | $^o$C |
| basal yield stress of ice ridge | $\tau_c$ | | kPa |
| ice stream thickness | $h_s$ | 900 | m |
| marginal inflow of ice | $q_r$ | $10^4$ | m$^2$ year$^{-1}$ |
| marginal lateral shear stress | $\tau_s$ | 200 | kPa |

## 3  Solution of the model

We solve the coupled mechanical and thermal system (2)–(17) with the finite element solver Elmer/Ice (Gagliardini et al., 2013). The computational domain is a relatively large, elongated rectangle which represents the margin cross-section in the $(y, z)$-plane. It consists of an ice and a sediment sub-domain. We apply the boundary conditions (5)–(6) and (14)–(16) at the relevant sides of the domain, rather than at $\pm\infty$.

The solutions to the problem are uniquely determined by the lateral shear stress $\tau_s$, the ice thickness $h_s$, marginal inflow of ice from the ridge $q_r$, the geothermal heat flux $q_{\text{geo}}$, and the surface temperature $T_s$, in addition to material properties such as thermal conductivity, heat capacity, density, rheological parameters for the ice, and the basal yield stress $\tau_c$. We will treat the majority of these material properties as fixed (see table 1), but consider carefully the effect of changing the basal yield stress.

## 3.1  Ice flow and heat production

We begin with solutions to the ice flow problem (2)–(10). In our model, we are treating viscosity and basal yield stress as independent from temperature. At present, this is necessary to allow the computation of more than a handful of solutions in a reasonable amount of time, mostly due to the difficulty involved in computing the migration rate from the inequality constraints

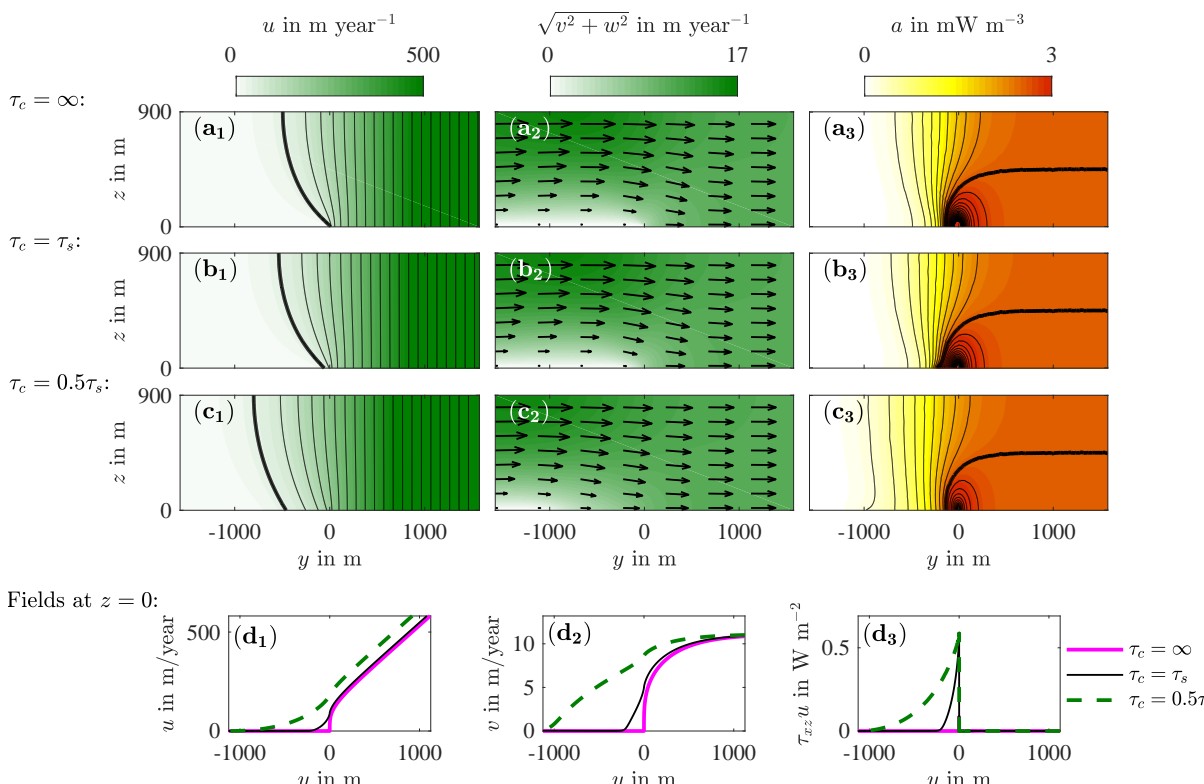

**Figure 2.** Influence of the subtemperate yield stress $\tau_c$ on the mechanical fields in the ice stream margin. $\tau_s$ is the lateral shear stress in the margin. Rows of panels are labeled $a$-$d$, with suffixes 1-3 indicating columns. Panels $a_1$–$c_1$: contours of downstream velocity component $u$ at contour intervals of 40 m/year; bold line indicates $u = 40$ m/year. Panels $a_2$–$c_2$: transverse velocity field $(v, w)$ shown as arrows, shading indicates magnitude of the transverse velocity. Panels $a_3$–$c_3$: contours of heat production $a$ at contour intervals $2.6 \times 10^{-4}$ W m$^{-3}$, $a = 2.56 \times 10^{-3}$ W m$^{-3}$ shown as a bold line. Panels $d_1$–$d_3$ show the velocities and heat dissipation at the bed. Note that no heat is dissipated at the bed where the velocities are zero. Calculations were done with the values listed in table 1 and $\tau_c$ as indicated.

(17). The latter requires very fine grids and a costly iterative scheme (Haseloff et al., 2015). We anticipate that future versions of the model will consider two-way coupling between the mechanical and thermal processes, but in our simplified version, we are able to compute solutions to the mechanical problem in isolation: given $h_s$, $\tau_s$, $q_r$, $\tau_c$ and the rheological properties $A$ and $n$, we are able to compute velocity and pressure in the ice.

5    The downstream velocity $u$ is vertically uniform in the ice stream far field and increases with a prescribed lateral gradient of $2A\tau_s^n$, see (5). Panels $a_1$–$c_1$ of figure 2 show contours of $u$ for different $\tau_c$ plotted in the $(y, z)$ plane, computed using the parameters in table 1. Panels $a_2$–$c_2$ show the corresponding across-flow field $(v, w)$, whose magnitude is significantly smaller than the downstream velocity. Hence gradients of the downstream velocity $u$ dominate the heat production rate, while the velocity $(v, w)$ in the transverse plane accounts for the advection.

In case of no slip on the cold side of the margin ($y < 0$), stress is concentrated around the transition from no slip to free slip at the origin. This translates to very high dissipation rates ($a$) as shown in figure 2, panel $a_3$. It can be shown that there is in fact a singularity with $a \sim 1/r$ in shear stress and consequently the heat production rate at the origin in this case (Rice, 1967, and supplementary material, section S3).

For decreasing $\tau_c$ (rows b and c of figure 2), the slip transition around the origin is smoothed, as a finite region with $y < 0$ forms where the yield stress $\tau_c$ is attained and sliding occurs. Panels $d_1$ and $d_2$ show the velocities $u$ and $v$ at the bed ($z = 0$). Allowing slip for $y < 0$ leads to two major changes in the heat production. First, the englacial heat production is reduced and the singularity at $y = 0$ is at least partially alleviated (see figure $2b_3$-$c_3$; the local analysis with $n = 1$ presented in the supplementary material, section S2 indicates that $a$ may still have a logarithmic singularity). Secondly, heat dissipation is

introduced along the ice–bed interface (see panel $d_3$ of figure 2). We will discuss below how this shift in the location of heat production affects the temperature field in the ice.

Panels $a_2$–$c_2$ of figure 2 show the transition of the transverse velocity component $v$ from a shearing flow to a plug flow. At the boundaries of the domain the vertical velocity component $w$ is zero. However, near the origin, we can observe a downwards motion of ice towards the bed, offsetting the accelerating transverse flow. As for $u$, allowing for subtemperate slip in a small

region at $y < 0$ leads to a more gradual increase in velocities around the origin (see panels $d_1$ and $d_2$ of figure 2).

## 3.2   Temperature field

To solve the heat equation (12), we need to know the migration rate $v_m$, which enters as a parameter. Without the inequality constraints (17a)$_1$–(17b)$_2$, any value of $v_m$ could be used to solve the heat equation. However, with an arbitrary choice of $v_m$, one or the other of these two inequalities is generally violated, which allows us to determine $v_m$ with an adapted bisection

method: the upper limit of the search interval is a migration velocity that is too big and therefore leads to a singular freezing rate on the ice stream side for $y > 0$, in violation of (17b)$_2$. The lower limit of the search interval leads to temperatures exceeding the melting point at the frozen side for $y < 0$, violating (17a)$_1$. At every iteration, we halve the search interval and continue the search in the upper half if (17a)$_1$ is violated at the midpoint and in the lower half otherwise (see supplementary section S1 for details).

In figure 3 we show temperature fields calculated with migration velocities that satisfy the inequality constraints (17a)–(17b). Each panel shows solutions obtained with a different combination of lateral shear stress $\tau_s$, lateral inflow of cold ice $q_r$ and basal yield stress $\tau_c$. Increasing $\tau_s$ while holding all the other parameters constant (top row) leads to more heating of the ice around the slip transition point and inside the ice stream. Consequently, the migration velocity increases. This gives the bed and the ice to the left of the transition point less time to heat up, and temperatures decrease there. The opposite effect can be

observed for an increase of the lateral inflow of ice (second row), which reduces the migration velocity and leads to a warmer bed. However, due to greater advection velocities, temperatures in the ice are lower (figure 3d–f).

Introducing subtemperate slip (decreasing $\tau_c$) leads to additional heat dissipation along the ice–bed interface on the ridge side ($y < 0$). This heat can thaw the frozen bed, thereby increasing the migration velocity. However, as in the case of increasing $\tau_s$ this gives the ice less time to heat up. Consequently, temperatures in the ice decrease as $\tau_c$ does.

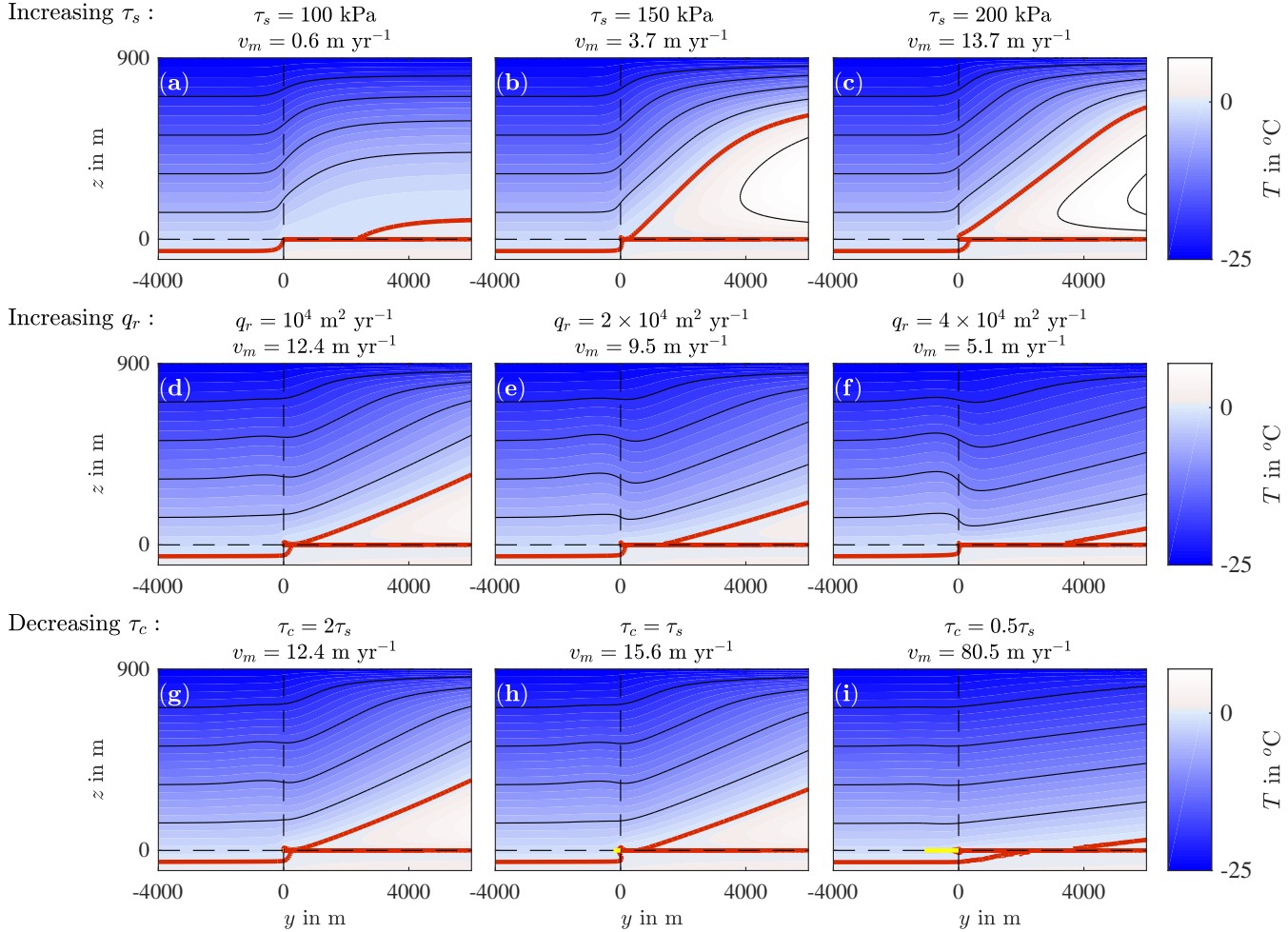

**Figure 3.** The effect of shear heating (represented by $\tau_s$), advection (represented by $q_r$), and subtemperate basal sliding (represented by $\tau_c$) on the temperature field in an ice stream margin, same plotting scheme as in figure 2. Solid black lines show contours in $5^o$C intervals with the contour of $T = 0^o$C in red. Thin dashed lines indicate the ice–bed interface and the location of the cold-temperate transition at the ice–bed interface. Red shading indicates the formation of temperate ice. Note that $T$ should be interpreted as a proxy for moisture content when $T > 0$. In this case we can identify $\phi = \rho c_p T/(\rho_w L)$ with $\phi$ the volumetric moisture content of the ice, $\rho_w$ the density of water, and $L$ latent heat per unit mass. Top row has $q_r = 0$, $\tau_c = \infty$ (i.e., no subtemperate slip), and values of $\tau_s$ as indicated above the panel. Middle row has $\tau_s = 200$ kPa and $\tau_c = \infty$. Bottom row has $\tau_s = 200$ kPa and $q_r = 10^4$ m$^2$ year$^{-1}$. The yellow lines in panels $g$ to $i$ mark the extent of the subtemperate slip region. All other values as listed in table 1.

Note that all the temperature fields shown in figure 3 have $T > 0$ in some parts of the ice. The boundaries of these regions are marked by a bold red line. In these regions of the ice stream, we solve the same heat equation as in the remainder of the domain (see also Schoof, 2012; Haseloff et al., 2015). Obviously, ice cannot have a temperature in excess of its melting point, and $T$ cannot be interpreted as temperature where $T > 0$. Effectively, we assume a very special case of an enthalpy gradient

model (Aschwanden et al., 2012; Schoof and Hewitt, 2016; Hewitt and Schoof, 2017): where $T > 0$, the product $\rho c_p T$ (which is generally the specific heat content per unit volume of ice) must instead be interpreted as the latent heat content per unit volume of the ice. That is, $\rho c_p T$ should be interpreted as $\rho_w L \phi$, where $\rho_w$ is the density of water, $L$ is latent heat per unit mass, and $\phi$ is the volumetric moisture content of the ice. This allows us to identify $\phi = \rho c_p T / (\rho_w L)$, so $T$ is nothing more than a proxy for moisture content when $T > 0$.

By solving the heat equation where $T > 0$, we make two main assumptions. First, qualitatively, we assume that moisture flows down gradients of moisture, which is the assumption common to enthalpy gradient models and permits the same diffusive model to be applied regardless of whether the melting point has been reached or not. The second, quantitative assumption we make is that the corresponding diffusivity remains the same for cold and temperate regions. This is consistent with prior work, but also an obvious area for future model improvement. We will return to a discussion of the limitations imposed by this assumption in section 5.

Importantly, the region of temperate ice in the bottom two rows of figure 3 does not form directly above the transition from subtemperate slip to free slip at the origin, but is shifted significantly (by up to several ice thicknesses) towards the ice stream. This is the result of lateral advection of ice and of subtemperate sliding, which generates additional heat and requires less localized englacial heating (compare the local form of the temperature field in appendix A and section S2 of the supplementary material with appendix A of Schoof (2012)). This shift of the temperate region away from the slip transition suggests that the thermal physics around the transition from frozen to unfrozen bed may be relatively unaffected by the choice of temperate ice model (e.g., Aschwanden et al., 2012; Schoof and Hewitt, 2016; Hewitt and Schoof, 2017).

## 4  Migration velocity as a function of forcing parameters

We now turn to a systematic investigation of the dependence of the migration velocity on the ice ridge and ice stream parameters. As we have pointed out, the solution to the velocity and temperature problem is determined uniquely once we know the applied lateral shear stress $\tau_s$, the inflow rate of cold ice $q_r$, the geothermal heat flux $q_{\text{geo}}$ (or equivalently, the far-field bed temperature on the ridge side of the margin $T_b$) as well as ice thickness $h_s$, basal yield strength $\tau_c$ and the remaining material properties of ice and bed. Importantly, that solution includes the margin migration rate $v_m$, which is therefore a function of these physical parameters and material properties: defining the far field basal temperature through

$$T_b = T_s + \frac{q_{\text{geo}} h_s}{k}$$

we can write

$$v_m = \hat{f}(h_s, q_r, \tau_c, \tau_s, T_b; A, c_p, c_{p,\text{bed}}, g, k, k_{\text{bed}}, n, \rho, \rho_{\text{bed}}, T_m, T_s). \tag{18}$$

We emphasize ice thickness $h_s$, inflow rate $q_r$, lateral shear stress $\tau_s$, and far-field bed temperature under the ridge $T_b$ in particular as these are parameters that reflect the coupling of the margin to dynamics of ice ridge and ice stream. Conceivably, one might want to run a simulation that relies on simplified models of ridge and stream without having to resolve the margin

region itself. The goal of a systematic solution of our margin model in that case is precisely to compute the migration rate $v_m$ as a function of parameters that are controlled by the ridge and the stream: doing so allows the margin to be treated as a free boundary in a larger-scale model. We also emphasize the role of basal yield stress $\tau_c$ as we are interested in how allowing for varying degrees of subtemperate sliding changes the relationship between margin migration rate and the forcing the margin

experiences from ridge and stream.

It is clear that $v_m$ in (18) depends on a large number of physical parameters, and the computational effort required to find the function appears to be intractable. However, we can reduce the parameter space to a minimum by non-dimensionalising the model: doing so demonstrates that many combinations of parameter values actually correspond to scaled versions of the same calculation, which we then have to do only once. This is done in section 4.1. An additional advantage is that non-

dimensionalisation allows us to identify systematically which processes dominate the temperature field and migration rate (see section 4.2). This leads to further simplification that allows us to give semi-analytical versions of (18) in a number of parameter regimes, which we study subsequently in sections 4.3–4.4.

## 4.1  Non-dimensionalisation

The goal of this section is to express the model in the most succinct form possible. To do so we introduce

$$[z] = h_s, \qquad [s'] = \frac{n+2}{n+1}\frac{q_r}{A\tau_s^n h_s^2}\frac{\tau_s}{\rho g}, \qquad [u] = A h_s \tau_s^n, \qquad [v] = \frac{n+2}{n+1}\frac{q_r}{h_s}, \qquad [T] = T_m - T_s \qquad (19)$$

and put $(y,z) = [z](Y,Z)$, $u = [u]U$, $(v,w) = [v](V,W)$, $s' = [s']S'$, $p = \rho g[s']P$, and $T = [T]\mathcal{T} + T_m$. This allows us to absorb quantities such as the ice thickness, inflow rate, and dimensionless lateral shear stress in the ice stream margin into five dimensionless parameters,

$$\alpha = \frac{A\tau_s^{n+1}h_s^2}{k(T_m - T_b)}, \qquad \mathrm{Pe} = \frac{(n+2)}{(n+1)}\frac{\rho c_p q_r}{k}, \qquad \nu = \frac{T_b - T_s}{T_m - T_s}, \qquad \tau = \frac{\tau_c}{\tau_s}, \qquad \varepsilon = \frac{n+2}{n+1}\frac{q_r}{A\tau_s^n h_s^2}. \qquad (20)$$

Note that our parameter $\alpha$ is defined slightly differently from its counterpart in Schoof (2012) and Haseloff et al. (2015): if we replace $T_m$ by $T_s$ in the denominator of $(20)_1$, we obtain the version of $\alpha$ used in the latter two papers. We can interpret the parameters above as a dimensionless shear heating rate $\alpha$, a Péclet number (or measure of advection versus conduction) $\mathrm{Pe}$, a dimensionless measure of the far field bed temperature $\nu$ ($\nu$ is between 0 and 1 for a ridge bed temperature $T_b$ below the melting point $T_m$, as we assume here), a dimensionless basal yield stress $\tau$, and a ratio of transverse to downstream velocities $\varepsilon$.

$\varepsilon$. Using the values in table 1, we get $\mathrm{Pe} = 314$, $\alpha = 592$, $\nu = 0.9$, $\varepsilon = 0.04$.

Note that a large Péclet number is what we would expect in a spatially confined region like an ice stream margin: conduction of heat is relatively ineffective, and advection mostly dominates. Large $\alpha$ reflects the strength of heat production, which must balance the fast rates of advection of cold ice implied by large $\mathrm{Pe}$. Note that $\varepsilon$ remains small as long as the across-margin flow is significantly smaller than the downstream flow, which we assume to be the case. Terms of $O(\varepsilon)$ are retained only in order

to regularize the viscosity in the ice ridge, where gradients in $u$ vanish. In the numerical solutions presented in this study, we use $\varepsilon = 0.01$, and we have confirmed that smaller values of $\varepsilon$ do not change our results. $O(1)$ values of $\varepsilon$ would imply that there is significant englacial heat production in the ridge, see Haseloff (2015) and Haseloff et al. (2015). This heat production

should prevent the ice ridge bed from remaining frozen, contradicting our basic assumption that the shear margin is co-located with a thermal transition at the bed. $\tau$ is poorly constrained, and we will consider different parameter regimes of $\tau$ below. Additionally, we have the following ratios of material properties

$$\gamma = \frac{\rho_{\text{bed}} c_{p,\text{bed}}}{\rho c_p}, \qquad \kappa = \frac{k_{\text{bed}}}{k}. \tag{21}$$

With the definitions above, the velocity in the downstream direction is determined by the scaled version of (2)

$$\frac{\partial}{\partial Y}\left(\mu \frac{\partial U}{\partial Y}\right) + \frac{\partial}{\partial Z}\left(\mu \frac{\partial U}{\partial Z}\right) = 0, \tag{22}$$

and the across-stream flow is described by (3)

$$\frac{\partial}{\partial Y}\left(2\mu \frac{\partial V}{\partial Y}\right) + \frac{\partial}{\partial Z}\left[\mu\left(\frac{\partial V}{\partial Z} + \frac{\partial W}{\partial Y}\right)\right] - \frac{\partial P}{\partial Y} = 0, \tag{23a}$$

$$\frac{\partial}{\partial Y}\left[\mu\left(\frac{\partial V}{\partial Z} + \frac{\partial W}{\partial Y}\right)\right] + \frac{\partial}{\partial Z}\left(2\mu \frac{\partial W}{\partial Z}\right) - \frac{\partial P}{\partial Z} = 0, \tag{23b}$$

$$\frac{\partial V}{\partial Y} + \frac{\partial W}{\partial Z} = 0. \tag{23c}$$

$\mu$ is the non-dimensional viscosity

$$\mu = \frac{1}{2^{1/n}}\left[\left|\frac{\partial U}{\partial Y}\right|^2 + \left|\frac{\partial U}{\partial Z}\right|^2 + \varepsilon^2\left(\left|\frac{\partial V}{\partial Z} + \frac{\partial W}{\partial Y}\right|^2 + 2\left|\frac{\partial V}{\partial Y}\right|^2 + 2\left|\frac{\partial W}{\partial Z}\right|^2\right)\right]^{\frac{1-n}{2n}}. \tag{24}$$

The boundary conditions in the ice stream far field (5) are now

$$\mu \frac{\partial U}{\partial Y} \to 1, \qquad \frac{\partial V}{\partial Z} \to 0, \qquad W \to 0 \quad \text{for} \quad Y \to \infty. \tag{25}$$

Towards the ice ridge, we obtain from (6)

$$U \to 0, \qquad V \to 1 - (1 - Z)^{n+1}, \qquad W \to 0 \quad \text{for} \quad Y \to -\infty. \tag{26}$$

At the ice surface, we have from the boundary conditions (10)

$$\mu \frac{\partial U}{\partial Z} = \mu \frac{\partial V}{\partial Z} = W = 0, \qquad 2\mu \frac{\partial W}{\partial Z} - P + S' = 0 \quad \text{at} \quad Z = 1. \tag{27}$$

As before, basal melting has a negligible effect on ice velocities, and (7) becomes

$$W = 0 \quad \text{at} \quad Z = 0. \tag{28}$$

On the temperate side of the bed, we have free slip from (8):

$$\mu \frac{\partial U}{\partial Z} = \mu \frac{\partial V}{\partial Z} = 0 \quad \text{at } Z = 0, \quad Y > 0. \tag{29}$$

On the frozen side of the bed, we can either have no slip (9a):

$$U = V = 0 \quad \text{at } Z = 0, \quad Y < 0, \tag{30a}$$

or we allow subtemperate slip, requiring from (9b)

$$\left.\begin{array}{ll} \text{either} & \mu\frac{\partial U}{\partial Z}=\tau\frac{U}{\sqrt{U^2+\varepsilon^2 V^2}}, \quad \mu\frac{\partial V}{\partial Z}=\tau\frac{V}{\sqrt{U^2+\varepsilon^2 V^2}}, \quad \sqrt{U^2+\varepsilon^2 V^2} \;>0 \\ \text{or} & \sqrt{\left(\mu\frac{\partial U}{\partial Z}\right)^2+\varepsilon^2\left(\mu\frac{\partial V}{\partial Z}\right)^2}\leq\tau, \quad \sqrt{U^2+\varepsilon^2 V^2} \;=0 \end{array}\right\} \text{ for } Y<0, \ Z=0. \tag{30b}$$

Note that the ice flow problem depends only on $n$, $\varepsilon$, and $\tau$.

For later convenience, we write the thermal problem in terms of a reduced temperature $\Theta$ through $\mathcal{T}=(1-\nu)\Theta-(1-\nu)-\nu Z$: $\Theta$ is the deviation from the linear temperature field that would result from geothermal heat flux and conduction alone, given the imposed surface boundary value. Writing the heat equation (12a)–(12b) in terms of $\Theta$ yields

$$V_m\frac{\partial\Theta}{\partial Y}+\mathrm{Pe}\left(V\frac{\partial\Theta}{\partial Y}+W\frac{\partial\Theta}{\partial Z}+\frac{\nu}{1-\nu}W\right)-\left(\frac{\partial^2\Theta}{\partial Y^2}+\frac{\partial^2\Theta}{\partial Z^2}\right)=\alpha\mathcal{A} \quad \text{for } 0<Z<1, \tag{31a}$$

$$V_m\gamma\frac{\partial\Theta}{\partial Y}-\kappa\left(\frac{\partial^2\Theta}{\partial Y^2}+\frac{\partial^2\Theta}{\partial Z^2}\right)=0 \quad \text{for } Z<0, \tag{31b}$$

with the heat production term

$$\mathcal{A}=\frac{1}{2^{1/n}}\left[\left|\frac{\partial U}{\partial Y}\right|^2+\left|\frac{\partial U}{\partial Z}\right|^2+\varepsilon^2\left(\left|\frac{\partial V}{\partial Z}+\frac{\partial W}{\partial Y}\right|^2+2\left|\frac{\partial V}{\partial Y}\right|^2+2\left|\frac{\partial W}{\partial Z}\right|^2\right)\right]^{\frac{1+n}{2n}} \tag{32}$$

where we have retained the small $O(\varepsilon^2)$ term, analogous to that in (24). We have also introduced $V_m$ as the dimensionless speed with which the ice stream margin migrates outwards. It is related to the dimensional migration speed through

$$v_m=\frac{k}{\rho c_p h_s}V_m. \tag{33}$$

The boundary conditions (14)–(16) are:

$$\Theta=0 \qquad \text{at } Z=1, \tag{34a}$$

$$\frac{\partial\Theta}{\partial Z}\to 0 \qquad \text{for } Z\to-\infty, \tag{34b}$$

$$\Theta\to 0 \qquad \text{for } Y\to-\infty, \tag{34c}$$

$$\frac{\partial\Theta}{\partial Y}\to 0 \qquad \text{for } Y\to\infty. \tag{34d}$$

Finally, we have the inequality constraints determining the migration velocity, which are

$$\Theta<1 \qquad \text{and} \qquad -\left.\frac{\partial\Theta}{\partial Z}\right|^+ +\kappa\left.\frac{\partial\Theta}{\partial Z}\right|^- =\begin{cases}0, & \text{if } \tau=\infty \\ \alpha\tau\sqrt{U^2+\varepsilon V^2}, & \text{if } \tau<\infty\end{cases} \quad \text{for } Y<0, \ Z=0, \tag{35a}$$

$$\Theta=1 \qquad \text{and} \qquad -\left.\frac{\partial\Theta}{\partial Z}\right|^+ +\kappa\left.\frac{\partial\Theta}{\partial Z}\right|^- \leq\begin{cases}0, & \text{if } \tau=\infty \\ \infty, & \text{if } \tau<\infty\end{cases} \quad \text{for } Y>0, \ Z=0. \tag{35b}$$

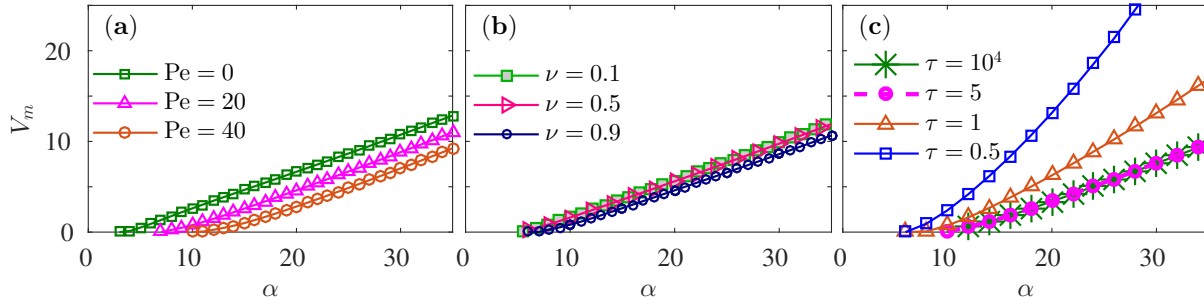

**Figure 4.** Dependence of non-dimensional migration velocity $V_m$ on lateral advection (parameterized by Pe, panel a), far-field bed temperature (parameterized by $\nu$, panel b) and basal shear stress on subtemperate side of the bed (parameterized by $\tau$, panel c). Unless indicated otherwise, we used $Pe = 10$, $\nu = 0.5$, and $\tau = \infty$.

We have now arrived at a model in which a (unique) dimensionless margin migration velocity $V_m$ is defined by four dimensionless groups that depend on forcing from the ice stream and ridge ($\alpha$, Pe, $\nu$, and $\tau$) and on the material constants $\gamma$, $\kappa$ and $n$:

$$V_m = f(\alpha, \mathrm{Pe}, \nu, \tau, n, \gamma, \kappa). \tag{36}$$

The remainder of this paper focuses on determining the form of the function $f$. For comparison with previous work (Schoof, 2012; Haseloff et al., 2015), we additionally assume $\gamma = \kappa = 1$ and $n = 3$, so that $V_m$ can only depend on Pe, $\alpha$, $\nu$, and $\tau$. We start by treating $\alpha$ and Pe as $O(1)$ parameters in the next section to build intuition for the dependence of $V_m$ on Pe, $\alpha$, $\nu$, and $\tau$ and then investigate the physically more realistic case in which both Pe and $\alpha$ are large in sections 4.3– 4.4.

### 4.2   Migration velocities at $O(1)$ values of $\alpha$ and Pe

In figure 4 we plot $V_m$ against $\alpha$ at fixed values of the other parameters. We see a qualitatively similar picture to the cases described in Schoof (2012) and Haseloff et al. (2015) which assumed a constant viscosity ($n = 1$): outward migration of the margin requires a minimum value of $\alpha$. Once that is reached, the migration speed increases with $\alpha$ (figure 4a). Increasing advection (Pe > 0) in the heat balance reduces the corresponding migration speed $V_m$. This is because heat production is now not only balanced by migration into the cold ice of the ridge, but also by the influx of cold ice into the boundary layer

(represented by Pe). In addition, for every fixed Péclet number there is a minimum nonzero value of $\alpha$ that generates a positive migration rate, as already found by Haseloff et al. (2015).

Solutions of $V_m$ for different $\nu$ between 0.1 and 0.9 are shown in figure 4b. Note that the migration velocity is not very sensitive to changes in $\nu$: for increasing $\nu$, there appears to be a small $\alpha$-independent shift of $V_m$ to smaller values. Consequently, relative differences in the migration rate should become smaller for increasing values of $\alpha$. If there are additional dependencies

of $V_m$ on $\nu$, these are small enough to be invisible. This behavior is consistent with the analysis for large $\alpha$ that we present later in section 4.3.

We have seen in the discussion of the mechanical fields in section 3.1 that subtemperate slip ($\tau < \infty$) introduces dissipation along the ice–bed interface. Decreasing $\tau$ leads to more subtemperate slip and therefore to more dissipation at the bed on the cold side of the margin. Consequently we expect the migration velocity to increase with decreasing $\tau$. Figure 4c confirms this. However, relatively small values of $\tau \lesssim 1$ are needed before there is a noticeable effect on the migration velocity.

A noticeable feature of figure 4 is not only that $V_m$ depends in qualitatively expected ways on $\alpha$, Pe and $\tau$. We also notice that the dependence of $V_m$ on $\alpha$ often appears to be nearly linear, and that $V_m$ is insensitive to changes in $\nu$. This suggests that, despite $f$ being a function of $\alpha$, Pe, $\tau$ and $\nu$, it may be possible to find parameter regimes in which simple representations of $f$ are available. In the remainder of the paper, we show that it is possible to derive such representations when the dimensionless heating rate $\alpha$ is large. Our estimates in section 4.1 have already indicated that this is the relevant regime that real ice stream margins should find themselves in. We begin by focusing on the case of no slip on the cold side, for which the relationship between $V_m$ and $\alpha$ in figure 4 appears to be nearly linear: we are able to demonstrate that this is the case, and give a formula for the resulting dependence on not only $\alpha$, but also Pe. Subsequently, we address the more complicated case of finite $\tau$, concluding with formulae for $V_m$ in the parameter regimes of large and moderate subtemperate slip.

### 4.3 Large heat production without subtemperate slip

We initially restrict ourselves to the case of no subtemperate slip, and consider the case of large $\alpha$: our estimates in section 4.1 indicate that this limit is likely to apply in practice. Combined with a large heat production rate, the same estimates lead us to expect a large Péclet number: ice is a relatively poor thermal conductor, and advection dominates conduction at the scale of a single ice thickness. Mathematically, this corresponds to advection and heating terms in the heat equation dominating over diffusion in (31) over most of the domain. However, this is no longer true close to the transition from no slip to free slip where there is a small region in which conduction also contributes to the local energy balance. The physics in this region determines the migration rate: conduction is an essential part of how the margin migrates, as it controls how heat production causes the cold part of the bed to warm, and how much heat is extracted from the temperate part of the bed. The analysis below therefore focuses on this small region (known technically as a 'conductive boundary layer', see figure 5a).

In what follows we give a brief description of how we can derive a model that ties migration velocity to heat production and transport in the conductive boundary layer. The reader not concerned with the technical details will find the result of this analysis in equation (43).

The non-dimensional mechanical problem (22)–(30b) is parameter-free in the absence of subtemperate slip (the $\tau = \infty$ case above). However, to analyze the temperature field in the boundary layer, we need to know the behavior of flow velocity and heat production near the transition from no slip to slip. In the supplementary section S3, we show that $\mathcal{A}$, $U$ and $(V, W)$ exhibit power-law behavior near the origin:

$$\left.\begin{aligned} &\mathcal{A} \sim A_\vartheta^{-1} R^{-1} \\ &U \sim \sqrt{\frac{2n}{n+1} A_\vartheta^{\frac{2}{n+1}} + \cos\vartheta\, A_\vartheta^{\frac{1-n}{1+n}} R^{\frac{1}{n+1}}} \\ &(V, W) \sim (V_\vartheta, W_\vartheta) R^\beta \end{aligned}\right\} \text{ as } R = \sqrt{Y^2 + Z^2} \to 0 \tag{37}$$

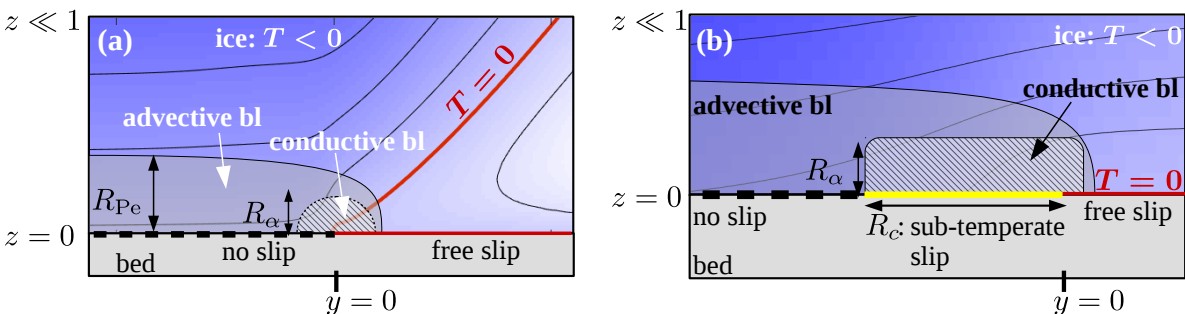

**Figure 5.** Boundary layer structure for asymptotics with large $\alpha$ and Pe, without subtemperate slip (panel $a$) and with subtemperate slip (panel $b$). The background temperature profiles are enlargements of typical profiles in this asymptotic limit, with same color scale as in figure 3. In the outer advective boundary layer the temperature field close to the bed is advected from the ice ridge towards the inner conductive boundary layer. In case of a no slip/free slip transition at the bed the conductive boundary layer consists of a small region around the slip transition (panel $a$). In case of subtemperate slip, the conductive boundary layer is a region of small vertical extent which stretches along the length of the subtemperate slip region (panel $b$). Within the conductive boundary layer, heat dissipation is balanced by diffusion.

with $\beta = 0.5$ for $n = 1$ and $\beta \approx 0.271$ for $n = 3$, and $A_\vartheta$, $V_\vartheta$, and $W_\vartheta$ functions of the angle $\vartheta$ between the vector $(Y, Z)$ and the $Y$–axis and independent of any other model parameters. Here, $R = \sqrt{Y^2 + Z^2}$ is distance from the origin. Knowledge of (37) enables us to study the behavior of the temperature field close to $R = 0$.

For large $\alpha$, heating near the origin behaves as $\alpha \mathcal{A} \sim \alpha R^{-1}$. As described above, we are looking for a region in which this heating rate is partially balanced by conduction. This happens at distances from the origin that scale as $R_\alpha = \alpha^{-1}$. To resolve this region, we set $(Y, Z) = R_\alpha(\widetilde{Y}, \widetilde{Z})$, $\mathcal{A} = R_\alpha^{-1}\widetilde{\mathcal{A}}$ and $(V, W) = R_\alpha{}^\beta(\widetilde{V}, \widetilde{W})$ using $(37)_1$ and $(37)_3$, and put $\widetilde{\Theta} = \Theta$. If the boundary layer sets the migration rate, then the effect of the margin migrating into colder ice must also enter into the energy balance of the boundary layer at leading order. In order for this to happen, we need a large migration velocity with $V_m \sim \alpha$, which we capture by rescaling the migration velocity as

$$\widetilde{V}_m = \alpha^{-1}V_m. \tag{38}$$

We can simultaneously consider conditions under which advection due to motion of the ice also contributes to the cooling of the conductive boundary layer, in addition to migration into cold ice. It turns out that this requires $\alpha^{-(1+\beta)}\mathrm{Pe}$ to be of $O(1)$, i.e., the Péclet number must scale as $\mathrm{Pe} \sim \alpha^{1+\beta}$. We therefore put

$$\Lambda = \mathrm{Pe}^{\frac{1}{1+\beta}}\frac{1}{\alpha} \tag{39}$$

and consider the case of $\Lambda \sim O(1)$ (known technically as a 'distinguished limit'). The case of slow advection is captured by taking the limit of small $\Lambda$. Fast advection of cold ice into the margin ($\Lambda \gg 1$) does not permit a widening of the ice stream as we are assuming here, and we exclude the case of large $\Lambda$ from consideration.

With these changes of variables and parameter definitions in place, (31) becomes (neglecting an $O(\alpha^{-1})$ term):

$$\widetilde{V}_m \frac{\partial \widetilde{\Theta}}{\partial \widetilde{Y}} + \Lambda^{(1+\beta)} \left( \widetilde{W} \frac{\partial \widetilde{\Theta}}{\partial \widetilde{Y}} + \widetilde{W} \frac{\partial \widetilde{\Theta}}{\partial \widetilde{Z}} \right) - \left( \frac{\partial^2 \widetilde{\Theta}}{\partial \widetilde{Y}^2} + \frac{\partial^2 \widetilde{\Theta}}{\partial \widetilde{Z}^2} \right) = \widetilde{\mathcal{A}} \quad \text{for } 0 < \widetilde{Z}, \tag{40a}$$

$$\widetilde{V}_m \frac{\partial \widetilde{\Theta}}{\partial \widetilde{Y}} - \left( \frac{\partial^2 \widetilde{\Theta}}{\partial \widetilde{Y}^2} + \frac{\partial^2 \widetilde{\Theta}}{\partial \widetilde{Z}^2} \right) = 0 \quad \text{for } \widetilde{Z} < 0, \tag{40b}$$

with boundary conditions given by the $\tau = \infty$ case in (35) and each variable being replaced by its rescaled version (i.e., $\Theta$ being replaced by $\widetilde{\Theta}$, $Y$ by $\widetilde{Y}$ and $Z$ by $\widetilde{Z}$). This boundary layer model contains only $\widetilde{V}_m$ and $\Lambda$ as parameters; if (as we expect) there is a unique migration velocity $\widetilde{V}_m$ which solves this problem for given $\Lambda$, then we have $\widetilde{V}_m = \widetilde{f}(\Lambda)$.

However, we are still missing conditions on $\widetilde{\Theta}$ at large distances from the origin, as we exit the conductive boundary layer and enter a region in which diffusion does not play the same leading order role (see figure 5a). These far-field conditions dictate how cold the ice that is advected into the conductive boundary layer is and therefore control in part the strength of conductive heat loss. In order to conclude that $\widetilde{V}_m$ depends on the parameters in the original scaled model (31) only through $\Lambda$, we need to be certain that these far-field conditions on $\widetilde{\Theta}$ also depend only on $\Lambda$. It turns out that these far-field conditions are determined by heat transport in a slender region near the bed. This region is marked with 'advective boundary layer' in figure 5a; it extends above the origin and towards the cold ice ridge. In this region, shear heating is balanced predominantly by advection of cold ice into the margin, and the effect of having to warm up ice towards the melting point as the margin migrates into the ridge. It can be shown that, at leading order, the heat equation in this region again contains only $\widetilde{V}_m$ and $\Lambda$ as parameters, and consequently, that the far-field conditions to (40) depend only on $\widetilde{V}_m$ and $\Lambda$ as required. This is somewhat tedious, and we give details in the supplementary material (section S4). Ultimately, we are able to confirm theoretically that

$$\widetilde{V}_m = \widetilde{f}(\Lambda). \tag{41}$$

Our goal now is to check numerically that this relationship is obtained from direct solutions of (22)–(35) when $\alpha$ and Pe are made sufficiently large, and to find the approximate form of the function $\widetilde{f}$. Note that we have gone from having a complicated function of 16 variables in equation (18) to being able to express the migration rate as a function of a single variable $\Lambda$ (which in turn depends on $\alpha$ and Pe). Approximating a function of a single variable numerically, for instance in the form of a look-up table, is obviously much simpler than having to solve numerically for a large number of independent variables, justifying the perhaps somewhat obscure procedure that has led us to this point (and its equivalent forms for other parameter regimes to follow later).

The limiting behavior (41) can also be written as

$$V_m = \alpha \widetilde{f} \left( \frac{\text{Pe}^{1/(1+\beta)}}{\alpha} \right).$$

Immediately, we see that for no advection (Pe $= 0$) we expect a linear relationship between migration rate $V_m$ and heating rate $\alpha$, which the results in Schoof (2012) and Haseloff et al. (2015) already hinted at for large $\alpha$. In fact, a linear relationship does not require vanishing Pe: it suffices that $\text{Pe}^{1/(1+\beta)}/\alpha \to 0$. We confirm this behavior numerically in figure 6a, where $\widetilde{V}_m$ is

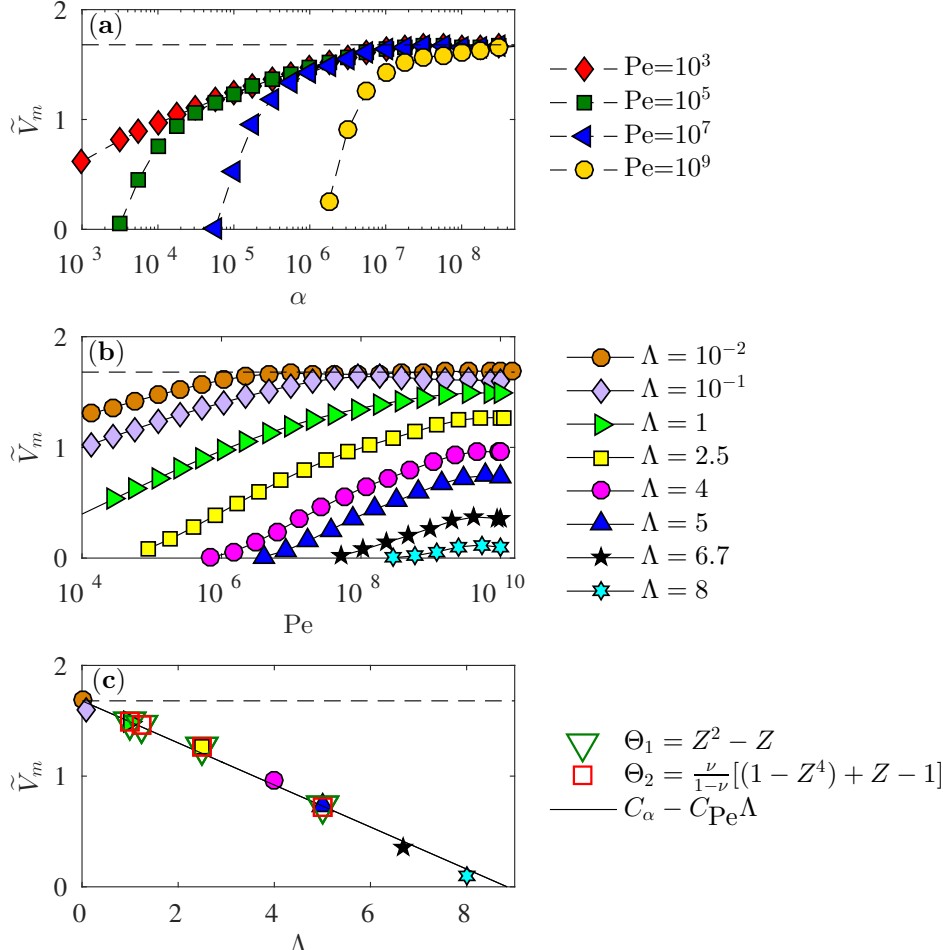

**Figure 6.** Asymptotic behavior of $\widetilde{V}_m = V_m \alpha^{-1}$ for the case without subtemperate slip. In this case we expect a limiting behavior of $\widetilde{V}_m = \widetilde{f}(\Lambda)$ for $\alpha \gg 1$ and $\Lambda = \mathrm{Pe}^{1/(1+\beta)}\alpha^{-1}$, see (41). Panel $a$: $\widetilde{V}_m$ against $\alpha$ for different values of constant Pe. Convergence to a constant value confirms the limiting behavior predicted by (41). Panel $b$: $\widetilde{V}_m$ against Pe for different values of $\Lambda$. Note that holding $\Lambda$ constant as Pe changes implies that $\alpha$ changes proportional to $\mathrm{Pe}^{1/(1+\beta)}$. Panel $c$: $\widetilde{V}_m$ against $\Lambda$ at constant $\mathrm{Pe} = 10^{10}$, corresponding to different values of $\widetilde{f}(\Lambda)$. Filled markers show the same data as in panel b. The open triangles and squares additionally show results for two different temperature fields in the ridge far field with same non-dimensional bed-temperature $\Theta = 0$ at $Z = 0$, as indicated in the legend.

plotted against $\alpha$ for different fixed values of Pe. $\widetilde{V}_m$ converges to approximately $1.68$ for each value of Pe, with the rate of convergence dependent on Pe (figure 6a).

A more general scenario is to consider $\alpha \to \infty$ and $\mathrm{Pe} \to \infty$ in such a way that $\Lambda$ is finite, in which case we cannot neglect the effect of advection. Figure 6b shows the convergence of $\widetilde{V}_m$ to its limiting form $f(\Lambda)$ as we make Pe (and hence $\alpha$) large while holding $\Lambda$ fixed. Note that holding $\Lambda$ fixed means that $\alpha$ must grow in lock-step with $\mathrm{Pe}^{1/(1+\beta)}$. The approach to the

limit can be relatively slow, though the limiting value gives a good order-of-magnitude estimate of the actual migration rate even for smaller values of Pe.

Finally, by plotting the converged values of $\widetilde{V}_m$ at large $Pe$ against $\Lambda$, we can find the function $\widetilde{f}(\Lambda)$ in (41). Figure 6c shows $\widetilde{V}_m$ plotted against $\Lambda$ for a fixed value of $\mathrm{Pe} = 10^{10}$, which is large enough for the limiting value to have been approached closely in all the examples shown in figure 6b. We can fit a linear relationship to the computational data, of the form

$$\widetilde{f}(\Lambda) = C_\alpha - C_{\mathrm{Pe}}\Lambda, \tag{42}$$

with $C_\alpha \approx 1.68$ and $C_{\mathrm{Pe}} \approx 0.19$, preserving the limiting value of $\widetilde{f}(0)$ identified above. Written in terms of the original migration velocity $V_m = \alpha\widetilde{f}(\Lambda)$, this is the same as

$$V_m = C_\alpha\alpha - C_{\mathrm{Pe}}\mathrm{Pe}^{\frac{1}{1+\beta}}, \tag{43}$$

for $\alpha \gg 1$ and $\mathrm{Pe} \gg 1$. As previously noted (see also Schoof, 2012; Haseloff et al., 2015), a finite heating rate $\alpha$ is required in order to cause outward migration of the margin, and the formula above is only valid for arguments $\alpha$ and Pe that ensure $V_m > 0$.

The migration rate in the limit of large $\alpha$ and Pe is set by heat generation and transport in a small conductive boundary layer near the no-slip-to-free-slip transition. As we have discussed above, the conductive boundary layer is subject to the advection of cold ice from the far field. That advection takes place from the ice ridge towards the margin, and crucially, the ice that eventually enters the conductive boundary layer always remains close to the bed. As a result, the conductive boundary layer is not sensitive to the details of the temperature profile with which ice enters the margin from the ridge, except for the basal temperature of ice in the ridge: ice at higher elevations simply passes over the boundary layer, and does not affect the energy balance that controls the migration rate at leading order. In technical terms, the far field boundary conditions on $\widetilde{\Theta}$ come from matching with the advective boundary layer alluded to above, which itself occupies only a small region near the bed and therefore has inflow boundary conditions dictated by the near-bed temperature prescribed in the limit $Y \to -\infty$, see the supplementary material, section S4 for details.

We can confirm computationally from solutions to (31) that $V_m$ is insensitive to the temperature profile imposed on the left-hand boundary. Still using large values of $\alpha$ and Pe, we solve (31) with the purely diffusive temperature profile prescribed in (34c) replaced by several nonlinear ones that have the same temperature at the bed as (34c), but with a steeper temperature gradient near the bed. The corresponding migration rates are displayed as open (empty) markers in figure 6c, and we find close agreement between results obtained from different far-field temperature profiles.

### 4.4 Large heat production with subtemperate slip

In the last section, we have considered large dissipation rates and rapid advection of ice, but no subtemperate slip. The migration velocity $V_m$ is then determined by heat production and transport in a small conductive layer around the no-slip-to-slip transition. The extent of that boundary layer scales as $R_\alpha = \alpha^{-1}$. Here, we extend the analysis for large $\alpha$ and Pe to account for subtemperate slip. When we allow for subtemperate slip in our model, sliding occurs on a patch of bed of finite size $R_c$, and the

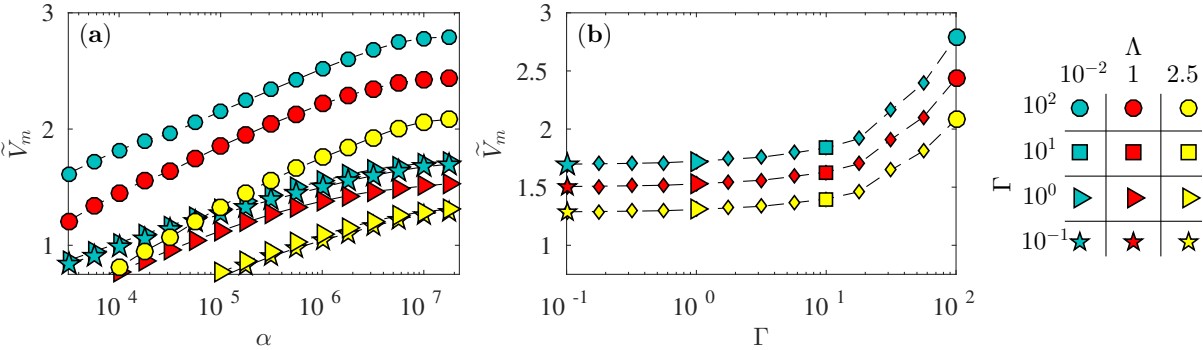

**Figure 7.** Panel $a$: asymptotic behavior for $\tau \gg 1$ where we expect a limiting behavior of $\widetilde{V}_m = \widetilde{g}(\Lambda, \Gamma)$ for $\alpha \gg 1$, see (46). Convergence to constant values confirms the limiting behavior. Note that $\alpha$ is plotted on a logarithmic scale. Panel $b$: velocities against $\Gamma = \tau^{-(n+1)}\alpha$ at constant values of $\Lambda = Pe^{1/(1+\beta)}\alpha^{-1}$ as indicated by color and $\alpha = 10^{7.25}$. Note that $\Gamma \to 0$ corresponds to $\tau \to \infty$, the limit without subtemperate slip.

size of that patch relative to the size of the diffusive boundary layer $R_\alpha$ becomes a key consideration (see figure 5b). We need to distinguish two basic cases, $R_\alpha \sim R_c$ and $R_c \sim O(1)$. We treat the former case first. Our modus operandi also remains the same as in the previous section: by rescaling the dimensionless temperature model (31) to capture the leading order behavior in the conductive boundary layer that determines the migration velocity, we derive a simplified form for the migration rate $V_m$ as

5 a function of the dimensionless parameters $\alpha$, Pe, $\nu$ and $\tau$, and test that relationship by solving (31) directly in the appropriate parameter regime.

### 4.4.1 A small slip region: $\tau \sim \alpha^{1/(n+1)} \gg 1$

Consider a slip region that is similar in size to the conductive boundary layer of section 4.3. To have such a small slip region, the dimensionless yield stress $\tau$ must be large. $\tau$ scales as $\tau \sim |\partial U/\partial Y|^{1/n}$ and with (37) we find that dimensionless stresses

10 in the ice scale as $R_\alpha^{-1/(n+1)} = \alpha^{1/(n+1)}$ in the conductive boundary layer of section 4.3. These must now be comparable to the dimensionless yield stress of the bed, hence

$$\tau \sim \alpha^{1/(n+1)}.$$

As we reduce $\tau$ from an effectively infinite value (so there is no slip, as in section 4.3) until it is comparable to $\alpha^{1/(n+1)}$, the magnitudes of velocity components and heat production rate in the conductive boundary layer remain the same as in

15 section 4.3, but the dependence of velocity and heat production on position starts to change, so the analytical formulae (37) no longer apply directly. (Technically, these formulae remain valid at distances $r$ from the origin for which $R_\alpha \ll r \ll 1$.) Knowing that the magnitudes remain the same, we can however use the same rescaling as in section 4.3, put $R_\alpha = \alpha^{-1}$ and set $(Y, Z) = R_\alpha(\widetilde{Y}, \widetilde{Z})$, $\mathcal{A} = R_\alpha^{-1}\widetilde{\mathcal{A}}$, $U = R_\alpha^{1/(n+1)}\widetilde{U}$, $(V, W) = R_\alpha^{\beta}(\widetilde{V}, \widetilde{W})$, and $\widetilde{\Theta} = \Theta$.

The resulting mechanical problem is detailed in the supplementary material, section S5. We do not go into detail here: the point is that the velocity $(\widetilde{U}, \widetilde{V}, \widetilde{W})$ and hence the heat production rate $\widetilde{\mathcal{A}}$ are fully determined if the ratio of yield stress $\tau$ to typical stress level $\alpha^{1/(n+1)}$ in the conductive boundary layer is given. We write that ratio (effectively, a dimensionless slip parameter) here in the form

$$\Gamma = \tau^{-(1+n)}\alpha. \tag{44}$$

Note that section 4.3 effectively treated the case of no slip, $\Gamma = 0$, and we seek to generalize this here. The corresponding problem for temperature $\widetilde{\Theta}$ again takes the form (40). In addition to $\widetilde{\mathcal{A}}$ now being dependent on the slip parameter $\Gamma$, the boundary conditions at the bed also depend on $\Gamma$: we have from (35a)–(35b) that

$$-\left.\frac{\partial\widetilde{\Theta}}{\partial\widetilde{Z}}\right|^{+} + \left.\frac{\partial\widetilde{\Theta}}{\partial\widetilde{Z}}\right|^{-} = \Gamma^{\frac{-1}{n+1}}|\widetilde{U}| \qquad\qquad \text{on } \widetilde{Z}=0, \widetilde{Y}<0 \tag{45a}$$

$$\widetilde{\Theta} = 1 \qquad\qquad \text{on } \widetilde{Z}=0, \widetilde{Y}>0 \tag{45b}$$

with the inequality constraints on flux and temperature still taking the same form as in (35a)–(35b).

In other words, the conductive boundary layer problem now depends on an additional parameter through $\Gamma$: with the abrupt transition from no slip to free slip (section 4.3), we had $\widetilde{V}_m = \widetilde{f}(\Lambda) \approx C_\alpha - C_{\mathrm{Pe}}\Lambda$ while now we have

$$\widetilde{V}_m = \widetilde{g}(\Lambda, \Gamma), \tag{46}$$

with $\widetilde{f}(\Lambda) = \widetilde{g}(\Lambda, 0)$. To confirm that (46) holds, we fix $\Lambda$ and $\Gamma$ to specific values and increase $\alpha$. This implies that $\mathrm{Pe} = (\alpha/\Lambda)^{1+\beta}$ and $\tau = (\alpha/\Gamma)^{1/(1+n)}$ both increase in lock-step with $\alpha$. Convergence of $\widetilde{V}_m$ to a value that depends only on $\Lambda$ and $\Gamma$ for $\alpha \to \infty$ then confirms (46).

Owing to the high computational cost of solving (31), especially in the limits of large $\mathrm{Pe}$ and $\alpha$ (when advection dominates and the conductive boundary layer requires high mesh resolution), we test for convergence of $\widetilde{V}_m$ in this parameter limit for a total of 39 combinations of $\Lambda$ and $\Gamma$, where we have restricted ourselves to only three values of $\Lambda$ and focused on the effect of changing the slip parameter $\Gamma$. Figure 7a shows the expected convergence in the limit of large $\alpha$. As in section 4.3, convergence often requires quite large values of $\alpha$. The limiting value of $\widetilde{V}_m$ is plotted against $\Gamma$ for the three different values of $\Lambda$ used in figure 7b.

As already observed in section 4.2, decreasing the basal shear stress increases the migration velocity due to increased dissipation on the cold side of the bed. We observe the same here, in the sense that increasing $\Gamma \propto \tau^{-(n+1)}$ increases the migration velocity. We also reproduce the limiting behavior for $\Gamma \to 0$ ($\tau \to \infty$), in which case we expect to reproduce the migration rate predicted for the no-slip-to-free-slip transition case of section 4.3. In fact, figure 7b shows that relatively large values of $\Gamma \approx 10$ are needed to see a significant departure of migration velocity $\widetilde{V}_m$ from its limiting value for $\Gamma = 0$. Unfortunately, computational constraints make it impossible for us to find a simple closed-form approximation for $\widetilde{g}$ analogous to (42): we simply do not have enough data to construct such an approximation. However, we will present a solution to this issue in section 4.4.3.

### 4.4.2 An $O(1)$ slip region: $\tau \sim O(1)$

We now turn to the case of $\tau \sim 1$, in which the lateral shear stress $\tau_s$ exerted by the ice stream on the margin is comparable with the yield strength $\tau_c$ of the frozen bed. In this limit, we expect the subtemperate slip length scale to be comparable with ice thickness, so $R_c \sim O(1)$ (see also figure 2). The region in which there is significant dissipation along the bed is now much
larger than in the previous section. As a result, the region in which dissipation is balanced substantially by conduction now has a horizontal extent comparable with ice thickness, too. For large $\alpha$, we however still have a conductive boundary layer whose vertical extent remains small: large temperature gradients are needed in order to account for the large amounts of dissipation, and such temperature gradients have to correspond to $O(1)$ temperature changes occurring over small vertical distances. In fact, that vertical distance still scales as $R_\alpha = \alpha^{-1}$. The primary difference is therefore that the boundary layer now has an
$O(1)$ extent in the horizontal, equal to the size of the slip region.

With an $O(1)$ region of slip at the bed, there are no simplifications to the mechanical problem (22)–(30b): we are no longer confining our attention to a small region around the origin. The solution to the mechanical problem is fully specified if we know $\tau$, so $(U, V, W)$ are functions of $\tau$ only. The horizontal velocity components $(U, V)$ are of $O(1)$. If we are concerned with the conductive boundary layer near the bed, then we only need the vertical velocity component near the bed. Since $W = 0$
at the bed itself, we find that $W \sim Z$ in the boundary layer. This allows us to rescale as $(Y, Z) = (Y^*, R_\alpha Z^*)$, $(U, V, W) = (U^*, V^*, R_\alpha W^*)$, $\mathcal{A} = \mathcal{A}^*$, and $\Theta = \Theta^*$. The leading order version of the heat equation (31) is thus (neglecting terms of $O(\alpha^{-1})$)

$$V_m^* \frac{\partial \Theta^*}{\partial Y^*} + \Omega \left( V^* \frac{\partial \Theta^*}{\partial Y^*} + W^* \frac{\partial \Theta^*}{\partial Z^*} \right) - \frac{\partial^2 \Theta^*}{\partial Z^{*2}} = 0 \tag{47a}$$

$$V_m^* \frac{\partial \Theta^*}{\partial Y^*} - \frac{\partial^2 \Theta^*}{\partial Z^{*2}} = 0 \tag{47b}$$

where we defined

$$V_m^* = \frac{V_m}{\alpha^2}, \qquad \Omega = \frac{\mathrm{Pe}}{\alpha^2} \tag{48}$$

and retained $\Omega$ as an $O(1)$ quantity: doing so with $\alpha \gg 1$ is again to look at a distinguished limit, analogous to treating $\Lambda$ as $O(1)$ in section 4.3. The rescaled version of the heat flux constraint (35a)$_2$ is

$$-\frac{\partial \Theta^*}{\partial Z^*} \bigg|^+ + \frac{\partial \Theta^*}{\partial Z^*} \bigg|^- = \tau |U^*|.$$

Again, there are far field boundary conditions on $\Theta^*$. These arise purely by advection into the boundary layer, and that advection occurs from the ice ridge, which fixes the far field temperature at $\Theta^* = 0$ as $Y^* \to -\infty$, so there is no additional parameter dependence through the far-field conditions. The thermal problem (47) contains only the dimensionless parameters $\Omega$ and $\tau$ (the latter both explicitly and through the velocity field). This indicates that the rescaled migration velocity $V_m^*$ only depends on $\Omega$ and $\tau$:

$$V_m^* = g^*(\Omega, \tau) \tag{49}$$

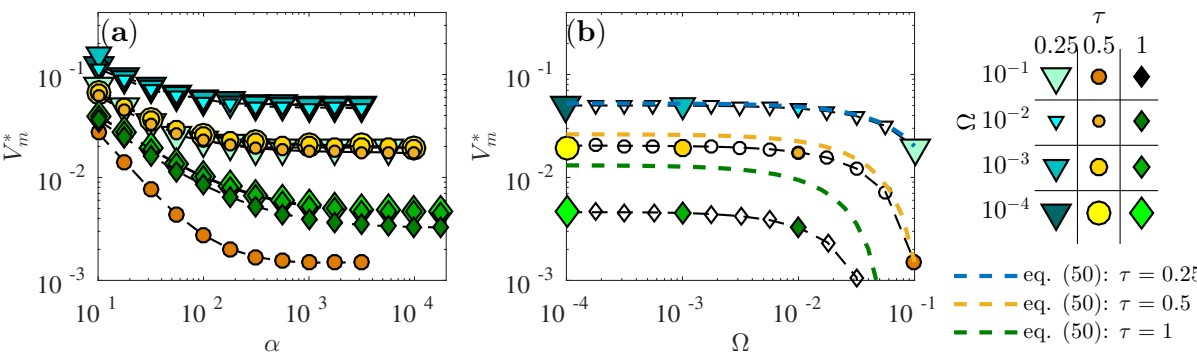

**Figure 8.** Panel $a$: asymptotic behavior for $\tau \sim 1$ where we expect a limiting behavior of $V_m^* = V_m \alpha^{-2} = g^*(\Omega, \tau)$ for $\alpha \gg 1$, see (49). The limiting behavior is confirmed by convergence to constant values for $\alpha \gg 1$. Panel $b$: asymptotic behavior of $V_m^*$ against $\Omega = \mathrm{Pe}\,\alpha^{-2}$ along constant values of $\tau$ at values of $\alpha$ where convergence is observed. Dashed lines show the analytical solution (50) which is valid for $\tau \ll 1$.

for $\alpha \gg 1$, $\mathrm{Pe} \sim \alpha^2$, and $\tau \sim 1$. Expressed in terms of the original migration velocity $V_m$, we have

$$V_m = \alpha^2 g^* \left( \frac{\mathrm{Pe}}{\alpha^2}, \tau \right).$$

Once more we confirm that the formula (49) holds by computing $V_m^*$ for different fixed $\Omega$ and $\tau$ while increasing $\alpha$. In this case, we can observe convergence at moderately large values of $\alpha \approx 10^2$ to $10^3$ (figure 8a). Again, smaller values of $\tau$ lead to larger values of $V_m^*$ and increasing $\mathrm{Pe}$ (and hence $\Omega$) leads to decreasing $V_m^*$ (figure 8b). Note that the solutions shown are computationally expensive: the conductive boundary layer invariably requires local mesh refinement, and in the parameter regime considered here, it extends over a larger part of the domain, with the size of the region that requires mesh refinement depending on $\tau$. As a result, we are not able to compute solutions for very large values of $\alpha$ at all values of $\tau$. Therefore, computational constraints once more mean that we are unable to sample a large enough region of the two-dimensional parameter space to give a simple formula for the function $g^*$.

However, for small values of $\tau \ll 1$, it is possible to solve the boundary layer problem analytically, as shown in the supplementary material, section S6. Effectively, this corresponds to finding the limiting behavior of $g^*$ as $\tau \to 0$, for which we obtain with $n = 3$ that

$$g^*(\Omega, \tau) \sim \frac{1}{\tau} \left[ \frac{64}{315\sqrt{\pi}} - \frac{63\sqrt{\pi}}{64} \Omega \tau \right]^2 \tag{50}$$

where the solution is valid only when the term in square brackets is positive: as before, a minimum value of $\alpha$ (equivalent to a maximum value of $\Omega \propto \alpha^{-2}$) is required to ensure outward migration of the margin. This limiting form is displayed in figure 8b along with the computationally obtained migration velocities for nominally small values of $\tau = 1$, $\tau = 0.5$ and $\tau = 0.25$ (note that $V_m \sim \tau^{-1}$, so $V_m$ does not approach a finite value, but diverges in a predictable fashion). For $\tau = 1$ and $\tau = 0.5$, there is still a notable difference between the numerical solution and the analytical solution (50), implying that these values of

$\tau$ do not yet satisfy $\tau \ll 1$. However, for $\tau = 0.25$, the difference between the numerical solution and the analytical solution is negligibly small.

### 4.4.3  Moderate slip: $1 \ll \tau \ll \alpha^{1/(1+n)}$

One of the difficulties we still face in making our results directly applicable to large scale models is that we have a closed-form approximation for the migration rate $V_m$ for only two parameter regimes. Both assume large dissipation rates $\alpha \gg 1$, which is realistic for abrupt ice stream margins. One applies to the case of no subtemperate slip (equation 43), while the other applies to extensive subtemperate slip (equation 50). The obstacle to dealing with more moderate amounts of subtemperate slip is that the functions $\widetilde{g}$ and $g^*$ in (46) and (49) are both functions of two independent variables, and that they are computationally expensive to evaluate.

There is one regime in which we can do better and reduce $\widetilde{g}$ and $g^*$ effectively to functions of a single variable: both functions have to be valid representations of the migration velocity $V_m$ in the limit $1 \ll \tau \ll \alpha^{1/(1+n)}$, in which $\Gamma = \tau^{-(1+n)}\alpha$ and $\tau$ are both large. In other words, in this parameter regime, we expect that $\widetilde{g}$ and $g^*$ give the same answer:

$$V_m \sim \alpha \widetilde{g}(\Lambda, \Gamma) \sim \alpha^2 g^*(\Omega, \tau).$$

If we denote the limiting forms of $\widetilde{g}$ and $g^*$ in the parameter limit under consideration with a subscript 0, then eliminating $\Omega$ and $\tau$ in favor of $\Gamma$, $\Lambda$ and $\alpha$ leads to

$$\alpha \widetilde{g}_0 (\Lambda, \Gamma) = g_0^* \left( \Lambda^{1+\beta} \alpha^{\beta-1}, \alpha^{1/(1+n)}\Gamma^{-1/(1+n)} \right).$$

By choosing Pe and $\tau$, we can vary $\Lambda$ and $\Gamma$ independently of $\alpha$. Hence, for any given $\Gamma$ and $\Lambda$, we can now pick $\alpha = \Gamma$ and are left with

$$\widetilde{g}_0 (\Lambda, \Gamma) = \Gamma^{-1} g_0^* \left( \Lambda^{1+\beta} \Gamma^{\beta-1}, 1 \right), \tag{51}$$

where we only need to evaluate $g_0^*$ at a fixed value of its first argument in order to compute $V_m/\Gamma$. Define $g(\chi) = g_0^*(\chi^{1-\beta}, 1)$ for arbitrary $\chi$. It then follows that we can use (51) and $V_m = \alpha \widetilde{g}_0 (\Lambda, \Gamma)$ to write

$$V_m = \tau^{-(1+n)}\alpha^2 g \left( \tau^{(1+n)} \mathrm{Pe}^{1/(1-\beta)} \alpha^{-2/(1-\beta)} \right). \tag{52}$$

As promised, the migration rate once more reduces to a function $g$ of a single variable.

Finding an approximation to $g$ requires significantly fewer function evaluations. Instead of varying $\tau$, Pe, and $\alpha$, we only need to vary the argument $\chi = \tau^{(1+n)} \mathrm{Pe}^{1/(1-\beta)} \alpha^{-2/(1-\beta)}$ of the function $g$. As before, we first confirm that (52) indeed holds by holding $\chi$ fixed and increasing $\alpha$ (figure 9a). Subsequently, we plot the limiting value of $V_m \tau^{1+n}\alpha^{-2}$ against $\chi$ in figure 9b. A simple polynomial fit

$$g(\chi) \approx \left[ c_2 \left( \chi - \chi_0 \right)^2 + c_4 \left( \chi - \chi_0 \right)^4 \right] \tag{53}$$

with $c_2 = 0.8$, $c_4 = 125$ and $\chi_0 = 0.07$ provides a good representation of the computed migration rates. The expression is again only valid for $0 < \chi \leq \chi_0$: the maximum allowed value of $\chi \propto \alpha^{-2/(1-\beta)}$ corresponds to the minimum dissipation rate $\alpha$ that ensures outward margin migration.

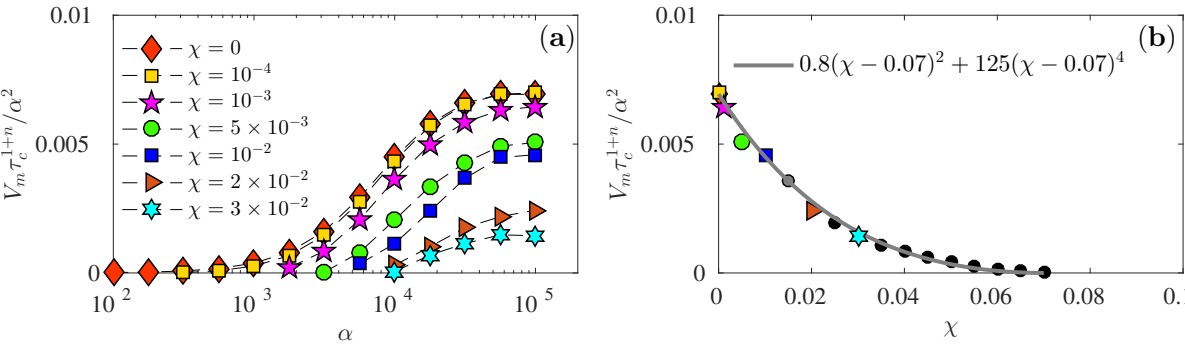

**Figure 9.** Asymptotic behavior for large $\alpha$ and Pe for the case of intermediate slip ($1 \ll \tau \lesssim \alpha^{1/(1+n)}$) where we expect a limiting behavior of $V_m \tau^{(1+n)} \alpha^{-2} = g(\chi)$, see (52). Panel $a$: $V_m \tau_c^{1+n}/\alpha^2$ against $\alpha$ for different values of $\chi = \tau_c^{(1+n)} \text{Pe}^{1/(1-\beta)} \alpha^{-2/(1-\beta)}$ with Pe fixed between 0 and $10^7$ and $\tau$ varying between 0.1 and 3.2. Convergence to constant values for $\alpha \gg 1$ confirms (52). Panel $b$: values of $V_m \tau_c^{1+n}/\alpha^2$ at different $\chi$ for $\alpha = 10^5$ and $\tau = 3.2$.

## 5 Discussion and Conclusions

In this study, we have investigated how different physical processes determine the widening of ice streams that are not topographically confined. We have considered the case in which the transition from fast to slow or no sliding that characterizes a typical ice stream margin is co-located with a thermal transition at the bed. In this scenario, the often intense dissipation of heat generated by the change in sliding behavior can cause the corresponding transition from a temperate to a cold bed to move, and the main objective of this study is to determine the corresponding rate of margin migration into the cold region. This ice stream widening relies on a delicate balance between heat dissipation, heat transport by advection, and conduction to warm the initially cold bed outside the ice stream. We have specifically excluded the case where heat loss dominates and the margin migrates into the ice stream from consideration here, although similar physics would allow inwards migration to be modeled (Schoof, 2012; Haseloff, 2015).

How the margin location is determined here differs from existing studies of heat transfer processes in Schoof (2004), Suckale et al. (2014), and Elsworth and Suckale (2016). In all of these, the mechanical transition and the thermal transition are not co-located. Instead, these studies appeal to spatial contrasts in basal friction caused by a heterogeneous drainage system as a mechanism for fixing the margin location independently of the thermal transition at the bed. These approaches and our approach likely represent end-members of how actual ice stream margins migrate, because hydrologically driven margin migration is excluded in our model: freezing in the bed leads to the formation of a thermal barrier which we assume subglacial water cannot penetrate (Haseloff, 2015). Observations suggest that the beds of ice ridges are indeed frozen in some parts (Bentley et al., 1998; Catania et al., 2003), but that widening of ice streams into these regions is nevertheless possible (Stephenson and Bindschadler, 1988; Fahnestock et al., 2000; Conway et al., 2002; Catania et al., 2012). It is therefore conceivable that

drainage-driven ice stream widening and thermally driven ice stream widening are operating in different regions of the Siple Coast ice streams and future work should investigate the interplay between these different processes.

To model the migration of ice stream margins, we solve a coupled model for ice flow and heat transport in the margin. In this model, the migration rate is determined by imposing constraints on the temperature and heat flux on the cold and warm side of the margin. The migration rate depends on material properties, ice geometry, lateral shear stress in the ice stream margin, and the velocity with which the ice enters the margin from the ridge. These dependencies can be expressed in terms of a small number of non-dimensional combinations of these parameters, although there is no closed form solution and the migration rate is expensive to evaluate on a case-by-case basis through the use of our model. In general, we have been able to establish that larger lateral shear stresses and less inflow of cold ice favor margin migration, as does a lower basal yield stress on the cold side of the margin.

To go further and provide quantitative parameterizations of the migration rate, we have exploited the fact that heat dissipation is generally large. This has allowed us to construct a number of approximate solutions for migration velocities that we can give in closed form. Where the different parameterizations we have derived apply depends on the amounts of subtemperate slip, controlled in our model by basal yield stress $\tau_c$. Note that all formulae given below are valid only where they predict $v_m > 0$: in general, there is a minimum value of dissipation rate required to produce any outward migration at all. In all cases, a Glen's law parameter $n = 3$ has been assumed.

For an infinite yield stress, which is equivalent to a sharp transition from no slip to free slip, the migration rate is (43):

$$v_m = 1.68 \frac{A\tau_s^4 h_s}{\rho c_p (T_m - T_b)} - 0.19 \frac{k}{\rho c_p h_s} \left( \frac{5}{4} \frac{\rho c_p q_r}{k} \right)^{0.79} \qquad \text{if } v_m \geq 0. \tag{54}$$

The parameter $A$ is the viscosity of ice, $\tau_s$ is the lateral shear stress in the ice stream margin, $h_s$ is the ice stream thickness, $q_r$ is the ice flux from the ice ridge into the ice stream, $T_b$ is the bed temperature in the ice ridge, $T_m$ is the melting point temperature, $c_p$ is the specific heat capacity of ice, $\rho$ is the density of ice, and $k$ is the thermal conductivity of ice. For an intermediate yield stress ($\tau_s \ll \tau_c < \infty$), we have from (53)

$$v_m = \frac{k}{\rho c_p h_s} \left( \frac{A\tau_s^4 h_s^2}{k(T_m - T_b)} \right)^2 \left( \frac{\tau_s}{\tau_c} \right)^4 \left[ 0.8 \left( \chi - 0.07 \right)^2 + 125 \left( \chi - 0.07 \right)^4 \right] \qquad \text{if } v_m \geq 0 \tag{55}$$

with $\chi$ given by

$$\chi = \left( \frac{\tau_c}{\tau_s} \right)^4 \left[ \frac{5}{4} \rho c_p q_r \frac{k(T_m - T_b)^2}{A^2 \tau_s^8 h_s^4} \right]^{1.4} ;$$

additionally, we must have $0 \leq \chi \leq 0.07$. The upper limit on $\chi$ corresponds to the lower limit on heat dissipation which is required for outwards migration. Finally, for a very small yield stress ($\tau_c \ll \tau_s$), we have equation (50)

$$v_m = \frac{k}{\rho c_p h_s} \left( \frac{A\tau_s^4 h_s^2}{k(T_m - T_b)} \right)^2 \frac{\tau_s}{\tau_c} \left[ \frac{64}{315\sqrt{\pi}} - \frac{315\sqrt{\pi}}{256} \frac{\rho c_p q_r}{k} \left( \frac{k(T_m - T_b)}{A\tau_s^4 h_s^2} \right)^2 \frac{\tau_c}{\tau_s} \right]^2 \qquad \text{if } v_m \geq 0. \tag{56}$$

There are also parameter regimes for which we cannot provide such succinct formulae, but in these regimes the migration rate still increases with $\tau_s$ and decreases with increasing lateral inflow of ice $q_r$.

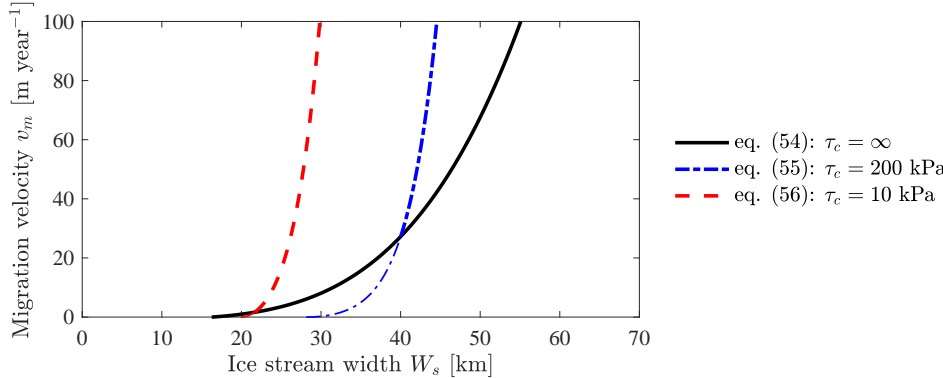

**Figure 10.** Margin migration velocities $v_m$ against ice stream width $W_s$. Equation (54) is valid for no subtemperate slip, $\tau_c = \infty$. Equation (55) is valid for intermediate values of the subtemperate yield stress, formally when $\tau_s \ll \tau_c \ll \tau_s^2[Ah_s^2/(kT_m - kT_b)]^{1/4}$. Equation (56) is valid for a small subtemperate yield stress (formally when $\tau_c \ll \tau_s$). Violation of the upper limit of validity of equation (55) leads to migration velocities that are less than the migration velocities without subtemperate slip (marked with thin blue line), which is unphysical, as subtemperate slip always increases migration velocities compared to no subtemperate slip.

Equations (54)–(56) describe margin migration as the result of a balance between englacial heat dissipation, heat dissipation along the ice–bed interface and advective cooling through the inflow of cold ice from the sides. The thermally active region where these processes operate is a small, conductive boundary layer around either the no slip/free slip boundary or along the region of subtemperate slip. While the migration velocity is determined by a balance between heat production and heat

transport processes on very small scales, equations (54)–(56) for $v_m$ only require knowledge of dynamic quantities that can be obtained in large scale ice sheet models ($\tau_s$, $h_s$, $q_r$, $T_b$, and $\tau_c$).

To illustrate how different geometric conditions alter the migration rate, we assume that the lateral shear stress is determined by $\tau_s = \rho g \sin \alpha W_s/2$ with $\sin \alpha = 10^{-3}$ the ice stream surface slope, and plot the migration velocity as a function of ice stream width $W_s$, figure 10. As one would expect, the migration velocity increases with increasing ice stream width $W_s$, because

wider ice streams are faster and therefore produce higher lateral shear stresses in the margin. All three parameterizations predict migration velocities that are of the order of the real-world migration velocities of 7 m/year to 30 m/year established from relatively sparse observations (Hamilton et al., 1998; Harrison et al., 1998; Echelmeyer and Harrison, 1999). Note that the curves representing solutions with subtemperate slip dip below the curve for the non-slip case. These parts of the solution curves corresponds to parameter regimes for which the approximate solutions are not expected to be valid: subtemperate sliding

always facilitates margin migration relative to the no-slip case.

Since the parameter ranges where equations (54)–(56) are applicable for a given ice stream depend on the yield strength of the subtemperate region, knowledge of the basal properties is necessary. There are only very few observations which would allow an estimate of the subtemperate basal yield strength (Holdsworth, 1974; Echelmeyer and Zhongxiang, 1987; Cuffey et al., 1999). Using the values reported in Cuffey et al. (1999) for Meserve Glacier, Antarctica, gives an approximate basal

yield stress of 380 kPa (neglecting the dependence on the thickness of the interfacial water layers). As the typical lateral shear

stress of an ice stream margin can be estimated from $\tau_s = \rho g \sin \alpha W_s / 2$ to be in the range of 180 kPa to 310 kPa (Joughin et al., 2002, table 3), this suggests that equation (55) will be appropriate in most circumstances.

Naturally, migration rate increases with ice stream width, hinting at the possibility of runaway widening of ice streams in a positive feedback. Note however that plotting $v_m$ against $W_s$ is somewhat misleading, as we have assumed the same ice thickness for different ice stream widths. In reality, we expect a negative feedback, where wider ice streams discharge more mass, thereby lowering the ice surface. This would lead to a decrease in englacial and subglacial dissipation, and slow the widening of the ice stream. Investigating this feedback requires combination of the parameterization of $v_m$ with a large-scale ice sheet model.

In principle, incorporating ice stream widening with equation (54)–(56) in large scale models should be possible; depending on the nature of the large scale model, the formula of the migration rate needs to be supplemented explicitly with the continuity conditions on ice thickness and lateral inflow of mass from Haseloff et al. (2015). There are some practical challenges, however: to calculate the migration velocity, the dynamic parameters $h_s$, $\tau_s$ and $q_r$ must be determined at the boundary between ice stream and ice ridge, which might not necessarily align with the mesh or grid of the ice sheet model. Additionally, the migration of this boundary at rates of a few meters per year will likely be below the mesh resolution of continental-scale models. In fact, high resolution is required to resolve ice streams to begin with. Use of equation (54)–(56) will therefore require methods that can adapt to moving ice stream boundaries, similar to methods for grounding lines of marine ice sheets (see e.g., Durand et al., 2009; Gladstone et al., 2012; Pattyn et al., 2012).

Moreover, we have only considered the evolution of an ice stream that is already fully evolved, but the physics governing the position of ice stream tributaries and ice stream inception remains unclear. The position of ice stream tributaries might be determined by geological factors (Peters et al., 2006), but it is conceivable that thermal transitions at the bed might play a role as well and that the pattern of ice stream tributaries is the result of an instability (Hindmarsh, 2009; Brinkerhoff and Johnson, 2015).

Our model as stated in section 2 makes several simplifying assumptions. We allow for subtemperate slip if the basal shear stress exceeds a constant yield stress $\tau_c$ to avoid the singular transition from no slip to free slip. However, an abrupt transition from a constant yield stress to free slip (corresponding to a zero yield stress) is still unlikely to occur in realistic situations. If subtemperate slip is facilitated by interfacial films, then the basal yield stress will depend on the temperature of the bed (Gilpin, 1979) and different temperature-dependent sliding laws have been suggested (Shreve, 1984; Fowler, 1986; Wolovick et al., 2014). Such a sliding law requires a two-way coupling between the solution of the mechanical model and the thermal model, potentially introducing feedbacks between these two. We will investigate this in a separate publication.

We also assume that the viscosity in the ice is independent of the ice temperature (i.e., $A = \text{constant}$). Recent studies of the velocity field and temperature in the margin of Whillans ice stream have found that matching observed profiles with numerical model results requires incorporating the temperature-dependence of viscosity (Suckale et al., 2014). However, in contrast to these studies, our boundary layer model is not intended to reproduce velocity and temperature profiles over the entirety of an ice stream. Instead, we focus on the processes in the ice stream margin that to leading order control margin migration. The asymptotic analysis of sections 4.2-4.4 would remain structurally the same if we accounted for a temperature-dependent

viscosity, though the form of the englacial dissipation term would change, as would, in most cases, the velocity. The most robust result in this regard is likely to be the $\tau \sim O(1)$ result of section 4.4, in particular the result (50) for small $\tau$ and a wide subtemperate slip region. For $\tau \sim O(1)$ or smaller, englacial dissipation affected by ice temperatures does not enter into the leading order basal energy balance that determines the margin migration rate, and the advection velocity that appears in that energy balance problem is set by flow at the ice thickness scale. In other words, the thermal boundary layer described by equation (47) is not changed by having a temperature-dependent viscosity. For a wide subtemperate slip region, where lateral flow takes the form of a plug flow, it then turns out that the derivation of equation (50) in the present paper (see section S6 of the supplementary material) remains the same for a temperature-dependent viscosity. As advection above the basal thermal boundary layer simply preserves the vertical temperature profile imposed by far field conditions in the ridge, the same depth-integrated calculation as in the supplementary material can be applied.

Finally, we have assumed that the dynamics of temperate ice can be represented by a particular version of an enthalpy gradient model (Aschwanden et al., 2012). We expect future iterations of our model to incorporate ice dynamics which can account for gravity-driven moisture transport in temperate ice (e.g., Schoof and Hewitt, 2016). A more sophisticated treatment of temperate ice is likely to be particularly relevant when temperate ice forms near the transition from a cold to a temperate bed: as we have seen, the Peclet number in a shear margin is large, and temperate ice formation down-flow from the cold-temperate transition is consequently unlikely to affect the temperature field close to the transition, as $T$ is dominated by advection. This makes our results with moderate to small $\tau_c$ (or, more accurately, with moderate to small $\tau$) likely to be the most robust to changes in the temperate ice model, since temperate ice forms some distance inside the ice stream in that case (see for instance fig. 3). We leave a deeper investigation to future work.

## Appendix A: Temperature close to the cold–temperate transition

Here we summarize the behavior of the velocity and temperature fields close a transition from frictional to free slip, based on calculations given in full detail in supplementary section S2. We assume a constant viscosity $\eta$, i.e., we assume $n = 1$, and we only consider the flow problem in the downstream direction (parallel to the margin), corresponding to very small distances from the origin, at which diffusion dominates heat transport, and deviations from the sliding velocity are small. In this case we can treat the velocity as the sum of a constant sliding velocity $\bar{u}_b$ at the transition from frictional to free slip, and a correction $\widetilde{u}(y, z)$, i.e., $u = \bar{u} + \widetilde{u}$. The correction velocity then satisfies for $z > 0$

$$\eta \nabla^2 \widetilde{u} = 0 \tag{A1}$$

with $\nabla$ the gradient operator in the transverse $y$-$z$-plane. $\widetilde{u}$ has to satisfy the boundary conditions

$$\eta \frac{\partial \widetilde{u}}{\partial z} = \begin{cases} \tau_c & \text{at } z = 0, y < 0 \\ 0 & \text{at } z = 0, y > 0. \end{cases} \tag{A2}$$

In the supplementary section S2, we show that this leads to a leading order heat dissipation term $a = \eta |\nabla \tilde{u}|^2 \sim \tau_c^2/(\pi^2 \eta)[\log(r/r_0)^2 + \vartheta^2]$ with $r$ and $\vartheta$ polar coordinates (i.e., $y = r\cos\vartheta$ and $z = r\sin\vartheta$). Note that the heat production rate has only a logarithmic singularity in the present case, while there is a $1/r$ singularity for a no-slip to free slip transition, see (37).

With the heat dissipation $a$ given, the temperature field close to the transition point satisfies at leading order

$$
\quad -k\nabla^2 T = \begin{cases} \frac{\tau_c^2}{\pi^2 \eta}[\log(r/r_0)^2 + \vartheta^2] & \text{for } z > 0 \\ 0 & \text{for } z < 0 \end{cases} \tag{A3}
$$

with the boundary conditions

$$
T(y,0) = 0 \quad \text{for } z = 0, y > 0 \tag{A4a}
$$

$$
-k\left[\frac{\partial T}{\partial z}\right]_-^+ = \tau_c \bar{u}_b \quad \text{and} \quad [T(y,0)]_-^+ = 0 \quad \text{for } z = 0, y < 0. \tag{A4b}
$$

In the ice ($0 < \vartheta < \pi$), the leading order solution of this is (see supplementary section S2)

$$
\quad T(r,\vartheta) = a_0 r^{1/2}\sin\left(\frac{\vartheta}{2}\right) + b_1 r\sin(\vartheta) + a_1 r^{3/2}\sin\left(\frac{3\vartheta}{2}\right) + b_2 r^2 \sin(2\vartheta) - \frac{\tau_c \bar{u}_b}{k} r\sin(\vartheta) + O\left(\frac{\tau_c^2}{4\pi^2 k\eta} r^2 \log(r)^2\right) \tag{A5a}
$$

and in the bed ($\pi < \vartheta < 2\pi$), the leading order solution is

$$
T(r,\vartheta) = a_0 r^{1/2}\sin\left(\frac{\vartheta}{2}\right) + b_1 r\sin(\vartheta) + a_1 r^{3/2}\sin\left(\frac{3\vartheta}{2}\right) + b_2 r^2 \sin(2\vartheta). \tag{A5b}
$$

The term $(\tau_c \bar{u}_b/k)r\sin(\vartheta)$ results from the interfacial heating along the ice–bed contact where subtemperate slip is possible. The $O(\tau_c^2/(4\pi^2 k\eta)r^2 \log(r)^2)$-term describes the contribution from the englacial heating, which is small in comparison to the contribution from subtemperate slip. Note that this is consistent with our results in section 4.4.2, where we found that the leading order heat equation (47) in the conductive boundary layer does not feature the englacial heat dissipation term, either.

If $a_0 \neq 0$, temperatures below the melting point for $y < 0$, $z = 0$ (on $\vartheta = \pi$) require $a_0 < 0$. However, this leads to a singular heat flux on the warm side ($z = 0$, $y > 0$)

$$
-k\left.\frac{\partial T}{\partial z}\right|_-^+ = -k\frac{1}{r}\left.\frac{\partial T}{\partial \vartheta}\right|_{\vartheta = 2\pi}^{\vartheta = 0} = -k\frac{a_0}{r^{1/2}} + O(1) \tag{A6}
$$

corresponding to a singular rate of freezing there. If we assume, as we do here, that a singular rate of freezing is not viable for a widening ice stream (see also Schoof, 2012), we must have $a_0 = 0$. With this choice, the temperature field at leading order is determined by the $O(r)$ terms:

$$
T(z) \sim b_1 z - \begin{cases} \frac{\tau_c \bar{u}_b}{k} z & \text{for } z > 0 \\ 0 & \text{for } z < 0. \end{cases} \tag{A7}
$$

Consequently, at leading order a finite net negative heat flux out of the bed $-k[\partial T/\partial z]_-^+ = \tau_c \bar{u}_b$ is possible for $y > 0$, corre-
sponding to finite (i.e., non-singular) freezing. This heat flux is independent of the across-stream coordinate $y$ and continuous

along $y$, in contrast to the case without subtemperate slip (see Schoof, 2012). Physically, this means that the existence of subtemperate slip on the cold side requires the removal of heat there and at the temperate side simultaneously. At the temperate side, where no heat is dissipated along the bed, this heat must be supplied in different form, most likely as latent heat transported by subglacial drainage.

5 *Competing interests.* The authors declare that they have no conflict of interest.

*Acknowledgements.* MH was supported by a Four Year Fellowship at the University of British Columbia, NSERC grant 357193-13, and the Princeton AOS Postdoctoral and Visiting Scientist Program. CS acknowledges NSERC grants 357193-13 and 446042-13. Numerical calculations performed on WestGrid facilities were supported by Compute Canada. We thank the editor Eric Larour and two anonymous referees for their thorough reviews which have helped to improve the manuscript.

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
