# Peer review of "The role of subtemperate slip in thermally-driven ice stream margin migration"

_The Cryosphere, 2017_

## Referee Comment (RC1) · Anonymous Referee #1 · 8 Apr 2018

General Comments:

The manuscript by Haseloff and coworkers addresses the enlargement of ice streams by the migration of margins that are controlled by a transition between temperate and subtemperate basal temperatures. Using a boundary layer approach, the analysis leads to predictions of outward migration rate that are formulated in terms of parameters that can be obtained from the behavior of the adjacent ice stream and ice ridge, with the ultimate goal of enabling the primary results of this study to be incorporated within large-scale ice sheet models. Several special cases are shown to admit simplified expressions that agree with detailed numerical results in particular limits that may be of relevance to different portions of the Siple Coast ice streams. The writing is clear and succinct, and the treatment and results are substantial and worthy of publication

without major revisions.

Specific Comments:

The analysis is very involved and appears to be technically sound. To further improve the utility of this work and its reception by the broader community, I strongly encourage minor revisions that are aimed at providing further context and physical insight. A dense mathematical treatment is no doubt warranted, but some brevity could be sacrificed to improve the communication of this substantial effort. For example, consider enlightening the audience as to why the focus is on ice stream enlargment rather than the more general case – (e.g., is it considered that a narrowing of ice stream width is less relevant or is it a harder problem for some reason?) the title does not reflect this one-sided focus. How were the particular parameter values listed at the bottom of table 1 chosen and where might they be considered relevant (or what ranges of values might be considered typical)? Figure 10 illustrates three different regimes of behavior, which is hepful, but some brief discussion of the likely importance of these different cases could be beneficial. The mathematics is interesting, but the physical insight it provides is what makes the effort worthwhile and this aspect of the paper could be improved.

In the heat balance treatment, it appears that the bed is ice free and so can warm from subtemperate to temperate conditions without latent heat effects. If the physical situation were different, I would have anticipated that this would strongly retard the rate of margin migration. Some further explanation should be provided for why only sensible heat need be considered beneath the ridge. Similarly, in the description surrounding Figure 3, I found the discussion surrounding the enthalpy gradient model unsatisfying. To be clear, are the contours in panels c, g, and h showing ice at +3C? Perhaps a brief discussion in the supplementary information might be used to outline why its okay once again to neglect latent heat.

What is it about Vm<0 that necessitates a different treatment strategy? I understand that there has been a deliberate decision to focus on cases with Vm>0, but it would be

helpful to give the reader some insight into why.

Technical Corrections:

Equation 10 doesn't need the :

The density of the bed in Table 1 must be wrong.

Line 22 on page 13 should read "constrained" not constraint

In many of the figures the symbols are so large as to obstruct the underlying lines (e.g. fig. 4). These could be shrunk or even eliminated in favor of combinations of dashes and dots.

The caption for figure 6c is confusing and the symbols for Theta1-3 are difficult to discern. This could be improved. As it stands, the description at the end of 4.3 describes results that are very difficult to see.

On line 32 of page 21 we are told that the stress scale can be found from (37), but further steps are required - maybe show this in the supplement.

On line 16 of page 28, give the physical explanation for the upper limit on chi - the number by itself is not particularly helpful.

---

## Referee Comment (RC2) · Anonymous Referee #2 · 4 May 2018

This paper derives a parameterization of the rate at which thermally-controlled shear margins migrate outwards based on several physical parameters that can be constrained observationally or theoretically. The model is based on a boundary-layer approximation published previously by the authors and the resulting governing equations are solved through the standard software package Elmer/Ice. The paper is well structured and clearly written.

To the best of my knowledge and understanding, the models and results presented here are novel, relevant, and mathematically and physically sound. The conclusions are thoroughly backed up by the results presented in the manuscript. There are no results that appear to be unsupported. I recommend this manuscript for publication after minor modifications have been made, as it would constitute a valuable contribution

to the community and readers of The Cryosphere.

That being said, I think there are a couple of opportunities for improving this manuscript that the authors should consider.

(1) A key goal of this paper is to propose a parameterization of the rate of margin migration that can be incorporated into large-scale ice-sheet models to capture this process. I like the ambition, but I think it would be valuable for the authors to state more explicitly that this parametrization is probably not generally applicable. There is the obvious issue of topographic controls, which are certainly relevant in some cases. Then, there is the assumption that the shear margin is thermally controlled and I think it would be useful for readers to learn whether the authors argue that they believe all margins are thermally controlled or whether they simply focus on this subset for the purposes of this study. Finally, the boundary-layer approach might not be equally applicable everywhere? For example, it is not clear to me that a margin that is wide as compared to the ice thickness scale (e.g., the Whillans ice plain with margins of 30km+) would be well described by this particular approximation.

(2) The boundary-layer model is very similar to the approach previously published by the same authors in JFM. One difference that stands out is the inclusion of subtemperate sliding leading to a different stress boundary condition. Are there other differences that I've missed? Given that the model approach as such has been published already, I'm not sure that it is necessary to introduce the model quite in this level of detail. Instead, I would suggest emphasizing the differences between the models more clearly. Given that the audience of TC is less theoretically inclined than the readers of JFM, I suggest a concise summary of the governing equations and boundary conditions together with a summary of the key assumptions behind the boundary-layer approach and the limitations entailed for applying this idea to field data. For example, the surface correction s' does not actually couple back to the mechanical model, so would this approach be suited to think about the Thwaites shear margin given the rapid thinning rates and differences in ice thickness in ridge/stream? In the discussion section, the

authors argue that they make two simplifying assumption, (1) assuming subtemperate slip and (2) ignoring temperature-dependence of stress and viscosity. I would argue that there are a lot more assumptions entailed in the model setup. I think it would be helpful to spell these out explicitly in the discussion.

(3) I was very surprised to read that the authors are unable to incorporate the temperature-dependence of the viscosity and the basal yield stress because of the large computational times apparently required to capture this. After all, this is "just" a 2D model? Ignoring the temperature-dependence of these two parameters strikes me as a very significant drawback of this analysis, both because both parameters are sensitively dependent on temperature (e.g., likely multiple orders of magnitude in the viscosity) and because both of these parameters are very consequential for margin migration. I am convinced that a more complete solution to this problem within reasonable computational time is well within the reach of modern numerics. That being said, I realize that switching the numerical approach would require an unreasonable amount of time by the authors, so I think the current route is acceptable. That being said, I suggest that the authors qualify the generality of their solutions very clearly and point out that ignoring temperature-dependence is likely not a minor detail.

---

## Author Comment (AC1) · 12 Jun 2018

**Response to reviewers:**

**The role of subtemperate slip in thermally-driven ice stream margin migration**

M. Haseloff, C. Schoof, O. Gagliardini

We thank both reviewers for their time and effort to review our manuscript. We have responded to the two reviews point-by-point below. We are listing reviewer comments in black, and respond to these comments in blue. Page, line, and equation numbers refer to the original manuscript unless indicated otherwise.

A marked-up version of our manuscript with tracked changes is attached to the end of this document.

**Anonymous Referee #1**

**General Comments:**

The manuscript by Haseloff and coworkers addresses the enlargement of ice streams by the migration of margins that are controlled by a transition between temperate and subtemperate basal temperatures. Using a boundary layer approach, the analysis leads to predictions of outward migration rate that are formulated in terms of parameters that can be obtained from the behavior of the adjacent ice stream and ice ridge, with the ultimate goal of enabling the primary results of this study to be incorporated within large-scale ice sheet models. Several special cases are shown to admit simplified expressions that agree with detailed numerical results in particular limits that may be of relevance to different portions of the Siple Coast ice streams. The writing is clear and succinct, and the treatment and results are substantial and worthy of publication without major revisions.

**Specific Comments:**

The analysis is very involved and appears to be technically sound. To further improve the utility of this work and its reception by the broader community, I strongly encourage minor revisions that are aimed at providing further context and physical insight. A dense mathematical treatment is no doubt warranted, but some brevity could be sacrificed to improve the communication of this substantial effort. For example, consider enlightening the audience as to why the focus is on ice stream enlargement rather than the more general case – (e.g., is it considered that a narrowing of ice stream width is less relevant or is it a harder problem for some reason?) the title does not reflect this one-sided focus. How were the particular parameter values listed at the bottom of table 1 chosen and where might they be considered relevant (or what ranges of values might be considered typical)? Figure 10 illustrates three different regimes of behavior, which is helpful, but some brief discussion of the likely importance of these different cases could be beneficial. The mathematics is interesting, but the physical insight it provides is what makes the effort worthwhile and this aspect of the paper could be improved.

We thank the reviewer for these suggestions. To address the question of narrowing versus widening, note that the physics is different. For widening, we have changed the following passage immediately before equation 8:

*"On the temperate (stream-ward) side of the ice–bed interface, we assume that the basal shear stress is negligible compared with the shear stresses in the ice, leading to a free slip boundary condition*: "

and to elaborate on this, we have introduced the following additional paragraphs immediately after equation 8:

*"In posing this boundary condition for an ice stream that is actively widening, we are assuming that an infinitesimal amount of melting of the bed suffices to allow for slip: once the thermal barrier at the bed is breached, we only need a very thin ice-free layer in order for slip to occur. This is consistent at least with the idea of a plastic bed, where slip can happen on a plane, or with a hard bed.*

*To the extent that additional degrees of freedom (other than temperature) are involved in sliding, the main concern would be presumably water pressure at the bed or within the till, rather than the thickness of the unfrozen till layer. Our assumption of a free slip once the melting point is reached is best justified (see Haseloff et al, 2015) if we suppose that the unfrozen bed is hydraulically well-connected, so that the water pressure in the parts of the bed that have just become unfrozen quickly equilibrates with water pressure elsewhere under the ice stream (and hence basal friction is comparable to the rest of the active ice stream). Shear stresses experienced by the margins of the ice stream are large compared with basal drag throughout the ice stream (Haseloff et al, 2015), and this implies that basal friction is small at leading order everywhere where the melting point is reached. There are undoubtedly other, more elaborate models for basal shear stress at the unfrozen bed; ours is the simplest possible case to analyse."*

The case of freezing is quite different, because here, the depth of the unfrozen till layer (if present) does matter: while sliding can potentially be activated by only melting out a small portion of the bed, shutting sliding down requires freezing all the way through any deformable till layer. We acknowledge this now in the following updated passage in the introduction

*"Under these conditions, the inwards migration (or narrowing) of an ice stream requires freezing of the entire sediment column (Schoof, 2012, appendix B). As melt water can be supplied to sections of the bed with active freezing from other regions of the ice stream via subglacial drainage, this necessitates taking into account the ice-stream wide energy balance and the thermal response of bed. Consequently, the inwards migration of ice streams is the result of insufficient heat dissipation over the width of the entire ice stream (Haseloff, 2015). However, as shown in Haseloff (2015) this process can at least in principle be modelled with large-scale ice sheet models without recourse to a boundary layer."*

In the paragraph that follows, we also expand on the difference between widening and narrowing:

*"In this scenario the outwards migration of ice stream margins requires melting of the frozen sediment under the ice ridge. By contrast with a narrowing ice stream, it is however not*

*necessary for the entire thickness of the sediment column to melt out: only part of it needs to be unfrozen to permit sliding, and we will later idealize this by assuming that sliding is possible as soon as the melting point is reached at the bed. This however also underlines the asymmetry between widening and narrowing of an ice stream, which motivates us to focus on the harder problem of widening, which requires heat to be transferred into the bed. Several studies show that a strong gradient in basal resistance created by a thermal transition at the bed leads to significant englacial heat production in the ice stream margins. "*

The values of the ice stream thickness $h_s$, marginal inflow $q_r$ and lateral shear stress $\tau_s$ are chosen to be representative of the upper margin of Whillans ice stream studied in Harrison et al (1998) and Echelmeyer & Harrison (1999) which the authors observed to migrate at a rate between 7 – 30 m/year. We have amended the caption of table 1 to clarify this.

To clarify which of the different parameterizations might be relevant where, we have added the following paragraph to the discussion section:

*"Since the parameter ranges where equations (54)-(56) are applicable for a given ice stream depends on the yield strength of the subtemperate region, knowledge of the basal properties is necessary. There are only very few observations which would allow an estimate of the subtemperate basal yield strength (Holdsworth, 1974; Echelmeyer and Zhongxiang (1987); Cuffey et al, 1999). Using the values reported in Cuffey et al. (1999) for Meserve Glacier, Antarctica, gives an approximate basal yield stress of 380 kPa (neglecting the dependence on the thickness of the interfacial water layers). As the typical lateral shear stress of an ice stream margin can be estimated from $\tau_s = \rho\, g\, \sin\alpha\, W_s/2$ to be in the range of 180 kPa to 310 kPa (Joughin et al, 2002, table 3), this suggests that equation (55) will be appropriate in most circumstances."*

In the heat balance treatment, it appears that the bed is ice free and so can warm from subtemperate to temperate conditions without latent heat effects. If the physical situation were different, I would have anticipated that this would strongly retard the rate of margin migration. Some further explanation should be provided for why only sensible heat need be considered beneath the ridge.

This is hopefully largely answered above.
To reiterate, we imagine the frozen part of bed to be either bed rock or an ice-sediment matrix, and therefore not necessarily ice free. The point is however that only a very small amount of the ice in the ice-sediment matrix (if present) needs to melt to create a layer that can support motion. We idealize this by saying that sliding becomes possible as soon as the melting point is reached, without requiring the full thickness of sediment to thaw out (similar physics is effectively assumed in e.g. Robel et al. (2014). Of course, latent heat effects will continue to enter into the energy balance of the bed: for our travelling wave formulation, this happens on the *warm* side, where the bed generally has a positive energy balance. That positive energy balance naturally goes into melting – and will melt out any remaining ice in the pore space of till (or equally, "dirty basal ice", which may be the same thing) before melting clean ice. This is an effect we could diagnose after the fact, but it does not enter into the dynamical part of our model.

Similarly, in the description surrounding Figure 3, I found the discussion surrounding the enthalpy gradient model unsatisfying. To be clear, are the contours in panels c, g, and h

showing ice at +3C? Perhaps a brief discussion in the supplementary information might be used to outline why it's okay once again to neglect latent heat.

This paper is probably not the place to explain the basics of enthalpy formulations, but we have expanded the text in the description of figure 3 (see below). The basic point is that latent heat is **not** neglected. Rather than interpreting T as temperature, it becomes necessary in regions where T > 0 to interpret ρ*c*T as the enthalpy content of the ice (i.e., the latent heat content), rather than as something proportional to actual temperature. This is a common approach in temperate ice modelling (see Aschwanden et al, 2012; Schoof and Hewitt, 2016; Hewitt and Schoof , 2017). The reason why this works is that ρ*c*T in the derivation of the heat equation starts out as internal energy (or enthalpy) content per unit volume, and the conservation law can be extended to cover latent heat as well.

By solving the heat equation in regions where the internal energy content comes in the form of latent rather than specific heat (the temperate regions, where our model computes T > 0), we make a very specific assumption about the flux of that energy: by retaining the heat equation, we are assuming that internal energy flows down its own spatial gradient, with a diffusivity that remains the same as when internal energy takes the form of specific heat (the "cold" part of the domain). The assumption that internal energy flows down its own gradient is the cornerstone of the widely used enthalpy gradient model as described at length in Aschwanden et al; our model goes a step further and assumes equal diffusivities in the cold and temperate regions. Neither assumption is really likely to be a good one (Schoof and Hewitt, 2016, Hewitt and Schoof, 2017).

In practice, because we mostly focus on high-Peclet-number situations with advection from the cold side, the effect of incorrectly modelling temperate ice is likely to be confined near the regions where our model predicts T > 0, and those regions are pushed to the right of the origin when we have subtemperate sliding. $V_m$ is ultimately controlled by the behaviour of the temperature field around the origin (where the two inequalities in (17) must both be satisfied within the any given neighbourhood of Y = 0). It is therefore likely that our treatment of temperate ice has little effect on the predicted $V_m$ in that situation (large Pe and moderate or small τ); for little or no subtemperate slip, a more careful look at temperate ice modelling is warranted, as we point out in section 5.

The text alteration we have made is the following, in section 3.2:

*"Note that all the temperature fields shown in figure 3 have T > 0 in some parts of the ice. The boundaries of these regions are marked by a bold red line. In these regions of the ice stream, we solve the same heat equation as in the remainder of the domain (see also Schoof, 2012, Haseloff et al, 2015). Obviously, ice cannot have a temperature in excess of its melting point, and T cannot be interpreted as temperature where T > 0. Effectively, we assume a very special case of an enthalpy gradient model (Aschwanden et al, 2012, see also Schoof and Hewitt, 2016, Hewitt and Schoof, 2017): where T > 0, the product ρ\*c_p\*T (which is generally the specific heat content per unit volume of ice) must instead be interpreted as the latent heat content per unit volume of the ice. That is, ρ\*c\*T should be interpreted as ρ_w\*L\*φ, where ρ_w is the density of water, L is latent heat per unit mass, and φ is the volumetric moisture content of the ice. This allows us to identify φ = ρ\*c_p/(ρ_w\*L)\*T, so T is nothing more than a proxy for moisture content when T > 0.*

*By solving the heat equation where $T > 0$, we make two main assumptions. First, qualitatively, we assume that moisture flows down gradients of moisture, which is the assumption common to enthalpy gradient models and permits the same diffusive model to be applied regardless of whether the melting point has been reached or not. The second, quantitative assumption we make is that the corresponding diffusivity remains the same for cold and temperate regions. This is consistent with prior work, but also an obvious area for future model improvement. We will return to a discussion of the limitations imposed by this assumption in Section 5."*

At the end of section 5, we have added:

*"Finally, we have assumed that the dynamics of temperate ice can be represented by a particular version of an enthalpy gradient model (Aschwanden et al., 2012). We expect future iterations of our model to incorporate ice dynamics which can account for gravity-driven moisture transport in temperate ice (e.g., Schoof and Hewitt, 2016). A more sophisticated treatment of temperate ice is likely to be particularly relevant when temperate ice forms near the transition from a cold to a temperate bed: as we have seen, the Peclet number in a shear margin is large, and temperate ice formation down-flow from the cold-temperate transition is consequently unlikely to affect the temperature field close to the transition, as $T$ is dominated by advection. This makes our results with moderate to small $\tau_c$ (or, more to the point, with moderate to small $\tau$) likely to be the most robust to changes in the temperate ice model, since temperate ice forms some distance inside the ice stream in that case (see for instance Fig. 3). We leave a deeper investigation to future work."*

What is it about $V_m < 0$ that necessitates a different treatment strategy? I understand that there has been a deliberate decision to focus on cases with $V_m > 0$, but it would be helpful to give the reader some insight into why.

This is hopefully covered by the above; to reiterate, we have included the following passages in the introduction

*"Under these conditions, the inwards migration (or narrowing) of an ice stream requires freezing of the entire sediment column (Schoof, 2012, appendix B). As melt water can be supplied to sections of the bed with active freezing from other regions of the ice stream via subglacial drainage, this necessitates taking into account the ice-stream-wide energy balance and the thermal response of bed. Consequently, the inwards migration of ice streams is the result of insufficient heat dissipation over the width of the entire ice stream (Haseloff, 2015). However, as shown in Haseloff (2015) this process can at least in principle be modelled with large-scale ice sheet models.*

*In this scenario the outwards migration of ice stream margins requires melting of the frozen sediment under the ice ridge. By contrast with a narrowing ice stream, it is however not necessary for the entire thickness of the sediment column to melt out: only part of it needs to be unfrozen to permit sliding, and we will later idealize this by assuming that sliding is possible as soon as the melting point is reached at the bed. This however also underlines the asymmetry between widening and narrowing of an ice stream, which motivates us to focus on the harder problem of widening, which requires heat to be transferred into the bed. Several studies show that a strong gradient in basal resistance created by a thermal transition at the bed leads to significant englacial heat production in the ice stream margins (...)."*

**Technical Corrections:**

Equation 10 doesn't need the :
Removed.

The density of the bed in Table 1 must be wrong.
Density, heat capacity, and thermal conductivity of the bed are set to the same values as in the ice, as we envision the bed as an ice/sediment mixture.

Line 22 on page 13 should read "constrained" not constraint
Corrected.

In many of the figures the symbols are so large as to obstruct the underlying lines (e.g. fig. 4). These could be shrunk or even eliminated in favor of combinations of dashes and dots.
The symbols are shrunk in figures 4, 6, 7, and 9 of the revised version of the manuscript.

The caption for figure 6c is confusing and the symbols for $\Theta_1$-$\Theta_3$ are difficult to discern. This could be improved. As it stands, the description at the end of 4.3 describes results that are very difficult to see.
The figure and caption have been updated.

On line 32 of page 21 we are told that the stress scale can be found from (37), but further steps are required - maybe show this in the supplement.
We have added extra information to the text to that should clarify how the stress scale can be found.

On line 16 of page 28, give the physical explanation for the upper limit on $\chi$ - the number by itself is not particularly helpful.
We have added the following statement to the text:

*"The upper limit on $\chi$ corresponds to the lower limit on heat dissipation which is required for outwards migration."*

**Anonymous Referee #2**

This paper derives a parameterization of the rate at which thermally-controlled shear margins migrate outwards based on several physical parameters that can be constrained observationally or theoretically. The model is based on a boundary-layer approximation published previously by the authors and the resulting governing equations are solved through the standard software package Elmer/Ice. The paper is well structured and clearly written.

To the best of my knowledge and understanding, the models and results presented here are novel, relevant, and mathematically and physically sound. The conclusions are thoroughly backed up by the results presented in the manuscript. There are no results that appear to be unsupported. I recommend this manuscript for publication after minor modifications have been made, as it would constitute a valuable contribution to the community and readers of The Cryosphere.

That being said, I think there are a couple of opportunities for improving this manuscript that the authors should consider.

(1) A key goal of this paper is to propose a parameterization of the rate of margin migration that can be incorporated into large-scale ice-sheet models to capture this process. I like the ambition, but I think it would be valuable for the authors to state more explicitly that this parametrization is probably not generally applicable. There is the obvious issue of topographic controls, which are certainly relevant in some cases. Then, there is the assumption that the shear margin is thermally controlled and I think it would be useful for readers to learn whether the authors argue that they believe all margins are thermally controlled or whether they simply focus on this subset for the purposes of this study. Finally, the boundary-layer approach might not be equally applicable everywhere? For example, it is not clear to me that a margin that is wide as compared to the ice thickness scale (e.g., the Whillans ice plain with margins of 30km+) would be well described by this particular approximation.

We thank the reviewer for these suggestions, which we have incorporated by making the following changes:

- To emphasize that we are only interested in ice streams that are not topographically controlled, we have changed the first sentence in the discussion to

  "*In this study, we have investigated how different physical processes determine the widening of ice streams that are not topographically confined.*"

- We have also added the following clarifying sentence to the model description after we describe the model geometry, which assumes a flat bed:

  "*Note however that this assumption does not exclude the application of our results to ice streams with a weak topographic control, as found in many regions of the Siple Coast: this assumption merely requires the elevation differences of the bed to be sufficiently small that it does not vary significantly over the lateral distance of a few ice thicknesses.*"

- A discussion of thermal vs hydrological controls is given in the second paragraph of the discussion/conclusion section, where we underline that we are interested in one

particular version of shear margins (and by implication admit that, for a particular margin, different physics might apply).

We do not think that most readers would identify an ice plain as an ice stream margin in the classical sense, especially if they look at the geometry in Figure 1, or if they see the paper in the context of prior literature (Raymond, 1996; Jacobson and Raymond, 1998; Schoof, 2006; Suckale et al., 2014; Perol and Rice, 2015). While the question of how such regions work mechanically is perfectly valid, we felt that it would actually be confusing to discuss them in the paper.

(2) The boundary-layer model is very similar to the approach previously published by the same authors in JFM. One difference that stands out is the inclusion of subtemperate sliding leading to a different stress boundary condition. Are there other differences that I've missed? Given that the model approach as such has been published already, I'm not sure that it is necessary to introduce the model quite in this level of detail. Instead, I would suggest emphasizing the differences between the models more clearly. Given that the audience of TC is less theoretically inclined than the readers of JFM, I suggest a concise summary of the governing equations and boundary conditions together with a summary of the key assumptions behind the boundary-layer approach and the limitations entailed for applying this idea to field data. For example, the surface correction s' does not actually couple back to the mechanical model, so would this approach be suited to think about the Thwaites shear margin given the rapid thinning rates and differences in ice thickness in ridge/stream? In the discussion section, the authors argue that they make two simplifying assumption, (1) assuming subtemperate slip and (2) ignoring temperature-dependence of stress and viscosity. I would argue that there are a lot more assumptions entailed in the model setup. I think it would be helpful to spell these out explicitly in the discussion.

The reviewer raises two main points, to which we respond separately:
1. The model should be written more concisely
2. The differences to the published paper in JFM should be made clearer, and the limitations of the model should be made clearer

Point 1:
Since reviewer #1 states above: "A dense mathematical treatment is no doubt warranted, but some brevity could be sacrificed to improve the communication of this substantial effort.", we have opted to not further shorten the model description, as this could make the paper even harder to read. In our collective experience publishing other modelling papers, truncated model descriptions usually lead to requests for a complete model (even if this has been stated in large part elsewhere).

Point 2:
We have added the following to the introduction:

*"The purpose of this paper is therefore twofold: (i) to use the margin boundary layer model of Haseloff et al. (2015) to investigate how subtemperate slip changes the heat production and temperature field in the ice stream margin, and thereby the rate at which ice streams can migrate outwards and (ii) to derive parameterizations of margin migration rate which can be used in large scale ice sheet models. Both of these points go beyond the work in Haseloff et al (2015); the parameterizations we derive in particular show how the limit of rapid advection of heat across the shear margin can be used to simplify the boundary layer*

*model and arrive at tractable forms of the migration rate that could be implemented in computational models either in the form of semi-analytical formulae or lookup tables.”*

We also have added further information about the derivation in section 2, to explicitly highlight that most simplifications of the model arise from the derivation of the boundary layer model given in Haseloff et al (2015), rather than from ad-hoc simplifications. For instance, we have added on page 3, line 23:

*“The asymptotic analysis in Haseloff et al (2015) shows that the boundary layer evolves rapidly in comparison to the ice stream and ice ridge, and is consequently quasi-static with the only time-dependence arising from the moving transition between a frozen and a temperate bed at $\pm y_m(x,t)$. Moreover, Haseloff et al (2015) show that the surface of the ice stream margin is flat at leading order and located at $z=h_s$, where $h_s$ is the ice thickness of the ice stream.“*

To answer the question about the surface correction *s'*, the fact that this does not couple back into the model but is simply computed diagnostically comes directly out of the asymptotic expansions in Haseloff et al (2015); this is once more not an ad hoc approximation but the natural leading order form of the problem. In order for this approximation (which appears to be common to previous work as well, such as Raymond 1996, Jacobson and Raymond 1998, Schoof 2004, Suckale et al 2014, all of whom use a flat upper surface for their computational domains) not to hold, an O(1) surface slope would have to exist in the shear margin – meaning, a surface angle comparable with 45 degrees. To our knowledge, such steep slopes do not usually occur at the edges of ice streams. (As with all asymptotics, our theory is never exact except in the limit, so we would expect to see deviations for moderate surface slopes, but the point of our work is clearly to get a tractable leading order result.)

With regard to the last point raised (“In the discussion section, the authors argue that they make two simplifying assumption, (1) assuming subtemperate slip and (2) ignoring temperature-dependence of stress and viscosity. I would argue that there are a lot more assumptions entailed in the model setup. I think it would be helpful to spell these out explicitly in the discussion.”), it would be helpful to know exactly which other simplifying assumption the referee has in mind. We chose to focus in the discussion on the physics that is not prescribed by the geometrical assumptions made in Haseloff et al (2015; long, narrow ice stream, ice stream width much greater than ice thickness) which gives most of the boundary conditions and the parallel-sided strip geometry of our boundary layer model. Two additional caveats we discuss are (3) the treatment of temperate ice (which we now discuss in greater detail at the very end of the discussion/conclusions section, and (4) the free slip boundary condition for Y > 0. For (3), we have added the following at the end of the paper:

*“Finally, we have assumed that the dynamics of temperate ice can be represented by a particular version of an enthalpy gradient model (Aschwanden et al., 2012). We expect future iterations of our model to incorporate ice dynamics which can account for gravity-driven moisture transport in temperate ice (e.g., Schoof and Hewitt, 2016). A more sophisticated treatment of temperate ice is likely to be particularly relevant when temperate ice forms near the transition from a cold to a temperate bed: as we have seen, the Peclet number in a shear margin is large, and temperate ice formation down-flow from the cold-temperate transition is consequently unlikely to affect the temperature field close to the transition, as T is dominated by advection. This makes our results with moderate to small $\tau_c$ (or, more to the point, with moderate to small $\tau$) likely to be the most robust to changes in the temperate ice model, since*

*temperate ice forms some distance inside the ice stream in that case (see for instance Fig. 3). We leave a deeper investigation to future work."*

In the modelling section, we have also expanded on our current formulation, adding

*"Note that all the temperature fields shown in figure 3 have $T > 0$ in some parts of the ice. The boundaries of these regions are marked by a bold red line. In these regions of the ice stream, we solve the same heat equation as in the remainder of the domain (see also Schoof, 2012, Haseloff et al, 2015). Obviously, ice cannot have a temperature in excess of its melting point, and T cannot be interpreted as temperature where $T > 0$. Effectively, we assume a very special case of an enthalpy gradient model (Aschwanden et al, 2012, see also Schoof and Hewitt, 2016, Hewitt and Schoof, 2017): where $T > 0$, the product $\rho*c_p*T$ (which is generally the specific heat content per unit volume of ice) must instead be interpreted as the latent heat content per unit volume of the ice,. That is, $\rho*c*T$ should be interpreted as $\rho_w*L*\phi$, where $\rho_w$ is the density of water, L is latent heat per unit mass, and $\phi$ is the volumetric moisture content of the ice. This allows us to identify $\phi = \rho*c_p/(\rho_w*L)*T$, so T is nothing more than a proxy for moisture content when $T > 0$.*

*By solving the heat equation where $T > 0$, we make two main assumptions. First, qualitatively, we assume that moisture flows down gradients of moisture, which is the assumption common to enthalpy gradient models and permits the same diffusive model to be applied regardless of whether the melting point has been reached or not. The second, quantitative assumption we make is that the corresponding diffusivity remains the same for cold and temperate regions. This is consistent with prior work, but also an obvious area for future model improvement. We will return to a discussion of the limitations imposed by this assumption in Section 5."*

With regard to point (4) (friction on the warm side $Y > 0$), we have added the following text in section 2:

*"In posing this boundary condition for an ice stream that is actively widening, we are assuming that an infinitesimal amount of melting of the bed suffices to allow for slip: once the thermal barrier at the bed is breached, we only need a very thin ice-free layer in order for slip to occur. This is consistent at least with the idea of a plastic bed, where slip can happen on a plane, or with a hard bed.*

*To the extent that additional degrees of freedom (other than temperature) are involved in sliding, the main concern would be presumably water pressure at the bed or within the till, rather than the thickness of the unfrozen till layer. Our assumption of a free slip once the melting point is reached is best justified (see Haseloff et al, 2015) if we suppose that the unfrozen bed is hydraulically well-connected, so that the water pressure in the parts of the bed that have just become unfrozen quickly equilibrates with water pressure elsewhere under the ice stream (and hence basal friction is comparable to the rest of the active ice stream). Shear stresses experienced by the margins of the ice stream are large compared with basal drag throughout the ice stream (Haseloff et al, 2015), and this implies that basal friction is small at leading order everywhere where the melting point is reached. There are undoubtedly other, more elaborate models for basal shear stress at the unfrozen bed; ours is the simplest possible case to analyse."*

(3) I was very surprised to read that the authors are unable to incorporate the temperature-dependence of the viscosity and the basal yield stress because of the large computational times apparently required to capture this. After all, this is "just" a 2D model? Ignoring the temperature-dependence of these two parameters strikes me as a very significant drawback of this analysis, both because both parameters are sensitively dependent on temperature (e.g., likely multiple orders of magnitude in the viscosity) and because both of these parameters are very consequential for margin migration. I am convinced that a more complete solution to this problem within reasonable computational time is well within the reach of modern numerics. That being said, I realize that switching the numerical approach would require an unreasonable amount of time by the authors, so I think the current route is acceptable. That being said, I suggest that the authors qualify the generality of their solutions very clearly and point out that ignoring temperature-dependence is likely not a minor detail.

We agree with the reviewer that in particular ignoring the temperature-dependence of the subtemperate yield stress is not a minor detail. However, in Haseloff et al (2015), figure 4a we show explicitly that a grid refinement to approximately $10^{-6}$ ice thicknesses is necessary to correctly model thermally-driven margin migration for O(1) parameters of $\alpha$ and Pe. Even further grid refinement is necessary to resolve the very small nested thermal boundary layers for the case of $\alpha \gg 1$ and Pe $\gg 1$, which required us to refine the grid down to $10^{-9}$ ice thicknesses. The necessity of this grid refinement has been confirmed with the analytic solutions provided in different asymptotic limits, which match the numerical solutions well. In the presence of subtemperate slip, this grid refinement has to be extended over the length of the subtemperate slip region, leading to ice-thickness wide regions with extremely high grid refinement. Additionally, each calculation of $V_m$ requires 20 or more individual calculations within the bisection method, as the migration velocity has to be found iteratively. All these requirements make the computational effort involved here significantly greater than for more conventional two-dimensional computational problems, in which the decoupling that we use to our advantage here would not be of great consequence.

Concerning the temperature-dependence of the viscosity, we have added the following text to the discussion:

*"However, our boundary layer model is not intended to reproduce velocity and temperature profiles over the entirety of an ice stream, but to focus on processes in the ice stream margin that to leading order control margin migration. The asymptotic analysis of sections 4.2-4.4 would remain structurally the same if we accounted for a temperature-dependent viscosity, though the form of the englacial dissipation term would change, as would, in most cases, the velocity. The most robust result in this regard is likely to be the $\tau \sim O(1)$ result of section 4.4, in particular the result (50) for small $\tau$ and a wide subtemperate slip region. For $\tau \sim O(1)$ or small, englacial dissipation affected by ice temperatures does not enter into the leading order basal energy balance that determines the margin migration rate, and the advection velocity that appears in that energy balance problem is set by flow at the ice thickness scale. In other words, the thermal boundary layer described by equation (47) is not changed by having a temperature-dependent viscosity. For a wide subtemperate slip region, where lateral flow takes the form of a plug flow, it then turns out that the derivation of equation (50) in the present paper (see section S6 of the supplementary material) remains the same for a temperature-dependent viscosity. Advection above the basal thermal boundary layer then simply preserves the vertical temperature profile imposed by far field conditions in the ridge, and the same depth-integrated calculation as in the supplementary material can be applied."*

[revised manuscript text omitted]